# A Unified Switching System Perspective and Convergence Analysis of Q-Learning Algorithms

**Donghwan Lee**
Korea Advanced Institute of Science and Technology
donghwan@kaist.ac.kr

**Niao He**
UIUC & ETH Zurich
niao.he@inf.ethz.ch

## Abstract

This paper develops a novel and unified framework to analyze the convergence of a large family of Q-learning algorithms from the switching system perspective. We show that the nonlinear ODE models associated with Q-learning and many of its variants can be naturally formulated as *affine switching systems*. Building on their asymptotic stability, we obtain a number of interesting results: (i) we provide a simple ODE analysis for the convergence of *asynchronous Q-learning* under relatively weak assumptions; (ii) we establish the first convergence analysis of the averaging Q-learning algorithm, and (iii) we derive a new sufficient condition for the convergence of Q-learning with linear function approximation.

## 1 Introduction

Reinforcement learning (RL) addresses the optimal control problem for unknown systems through experiences [30]. Q-learning, originally introduced by Watkins [36], is one of the most popular and fundamental reinforcement learning algorithms for unknown systems described by Markov decision processes. The convergence of Q-learning has been extensively studied in the literature and proven via several different approaches, including the original proof [36], the stochastic approximation and contraction mapping-based approach [14, 33, 2, 9, 32, 9, 1, 38], and the ODE (ordinary differential equation) approach [4].

The ODE approach analyzes the convergence of general stochastic recursions by examining stability of the associated ODE model [3, 17, 4] and has been used as a convenient analysis tool to prove convergence of many RL algorithms, especially the temporal difference (TD) learning algorithm [29] and its variants [23, 31, 19, 10]. However, its application to Q-learning has been limited due to the presence of the max-operator, which makes the associated ODE model a complex nonlinear system. In contrast, the associated ODE of TD learning for policy evaluation is a linear system, whose asymptotic stability is much easier to analyze in general. While [4] gave the convergence proof of Q-learning based on a nonlinear ODE model, to the authors' knowledge, substantial analysis is required to prove the stability of the corresponding nonlinear ODE [5] by using the max-norm contraction of the Bellman operator. In addition, the result in [4] only applies to synchronous Q-learning, where every state-action pair is visited at each iteration, instead of the commonly used asynchronous Q-learning. Last but not least, the stability analysis does not immediately extend to other Q-learning variants, such as double Q-learning [11], averaging Q-learning [19], and Q-learning with linear function approximation, etc.

In this paper, we provide a simple and unified framework to analyze Q-learning and its variants through switched linear system (SLS) models [21] of the associated ODE. SLSs are an important class of nonlinear hybrid systems, where the system dynamics matrix switches within a finite set of subsystem matrices (or modes) according to a switching signal. The study of SLSs has attracted tremendous attention in the past years and their stability behaviors have been well established in the

literature; see [22] and [21] for comprehensive surveys. Our main contributions are summarized as follows:

1. For a number of Q-learning algorithms such as the asynchronous Q-learning, we show that the nonlinear ODE models associated with these algorithms can be characterized as affine switching systems with a state-feedback switching policy.

2. We construct both upper and lower comparison systems of the corresponding affine switching systems, and prove their asymptotic stability based on existing control theory and comparison principles. As a result of the Borkar and Meyn theorem [4], we obtain the asymptotic convergence of these Q-learning algorithms.

3. We extend the approach to analyze the averaging Q-learning [19]. To our best knowledge, this is the first convergence analysis of averaging Q-learning in the literature.

4. We also examine Q-learning with linear function approximation and derive a new sufficient condition to ensure its convergence based on the switching system theory. We show that, under specific assumptions, our new diagonal dominating condition is weaker than the well-known Melo's condition provided in [24].

**Related Work.** There exists few work on the non-asymptotic convergence rate of these classical Q-learning algorithms such as synchronous Q-learning [34, 9], asynchronous Q-learning [32, 9, 27], Q-learning with linear function approximation [6], etc. Most of the analyses build on completely different techniques and whether these finite-time bounds are sharp or not remains an open question. On the other hand, there is growing interest on designing variants of Q-learning algorithms with improved performance guarantees, e.g., [8, 1, 20, 15, 18], to name a few. Different from these lines of work, the goal of this paper is to establish an initial connection between switching systems and a family of Q-learning algorithms and provide a unified convergence analysis technique. This could potentially open up new opportunities to the development of a tight non-asymptomatic analysis for Q-learning algorithms and the design of new RL algorithms.

It is worth mentioning that several recent work established the analysis of reinforcement learning algorithms based on their connections to control theory. For example, [28] provided the finite sample bound of TD learning based on Lyapunov stability theory for linear ODE. [6] extended the analysis to Q-learning with linear function approximation. [13] explored the connection between temporal difference (TD) learning and the Markov jump linear systems (MJLS). Note that MJLS cannot be used to characterize the nonlinear dynamics of Q-learning. Instead, we resort to linear switching systems with state-feedback switching policies. Our new ODE approach based on linear switching systems can be used as a viable alternative to prove the stability of the associated ODE of various reinforcement learning algorithms as well as their asymptotic (and potentially non-asymptotic) convergence. Finally, we remark that an earlier work [23] also exploited the stability of linear switching system for Greedy-$GQ$ to establish the boundedness of iterates.

## 2 Preliminaries: MDPs, switching systems, and stochastic approximation

### 2.1 Markov decision problem

We consider the infinite-horizon (discounted) Markov decision process (MDP) with state-space $\mathcal{S} := \{1, 2, \ldots, |\mathcal{S}|\}$, action-space $\mathcal{A} := \{1, 2, \ldots, |\mathcal{A}|\}$, transition matrices $P_a \in \mathbb{R}^{|\mathcal{S}| \times |\mathcal{S}|}, a \in \mathcal{A}$, where $P_a(s, s')$ is the probability transiting from state $s$ to the next state $s'$ under action $a \in \mathcal{A}$, and random reward function $r_a(s, s')$. A deterministic policy, $\pi : \mathcal{S} \to \mathcal{A}$, maps a state $s \in \mathcal{S}$ to an action $\pi(s) \in \mathcal{A}$. The goal is to find a deterministic optimal policy, $\pi^*$, such that the cumulative discounted rewards over infinite time horizons is maximized, i.e., $\pi^* := \arg\max_{\pi \in \Theta} \mathbb{E}\left[\sum_{k=0}^{\infty} \alpha^k r_{a_k}(s_k, s_{k+1}) \big| \pi\right]$, where $\gamma \in [0, 1)$ is the discount factor, $\Theta$ is the set of all admissible deterministic policies, $(s_0, a_0, s_1, a_1, \ldots)$ is a state-action trajectory generated by the Markov chain under policy $\pi$. The Q-function under policy $\pi$ is defined as

$$Q^{\pi}(s, a) = \mathbb{E}\left[\sum_{k=0}^{\infty} \gamma^k r_{a_k}(s_k, s_{k+1}) \Bigg| s_0 = s, a_0 = a, \pi\right], \quad s \in \mathcal{S}, a \in \mathcal{A},$$

and the corresponding optimal Q-function is defined as $Q^*(s, a) = Q^{\pi^*}(s, a)$ for all $s \in \mathcal{S}, a \in \mathcal{A}$. Once $Q^*$ is known, then an optimal policy can be retrieved by $\pi^*(s) = \arg\max_{a \in \mathcal{A}} Q^*(s, a)$.

## 2.2 Basics of nonlinear system theory

Consider the nonlinear system

$$\frac{d}{dt}x_t = f(x_t), \quad x_0 = z, \quad t \in \mathbb{R}_+, \tag{1}$$

where $x_t \in \mathbb{R}^n$ is the state and $f : \mathbb{R}^n \to \mathbb{R}^n$ is a nonlinear mapping. For simplicity, we assume that the solution to (1) exists and is unique. This holds true if $f$ is globally Lipschitz continuous.

**Lemma 1** ([16, Theorem 3.2]). *Consider the nonlinear system* (1) *and assume that $f$ is globally Lipschitz continuous, i.e., $\|f(x) - f(y)\| \le L\|x - y\|$, $\forall x, y \in \mathbb{R}^n$, for some $L > 0$ and norm $\|\cdot\|$, then it has a unique solution $x(t)$ for all $t \ge 0$ and $x_0 \in \mathbb{R}^n$.*

An important concept in dealing with the nonlinear system is the equilibrium point. A point $x = x^e$ in the state space is said to be an equilibrium point of (1) if it has the property that whenever the state of the system starts at $x^e$, it will remain at $x^e$ [16]. For (1), the equilibrium points are the real roots of the equation $f(x) = 0$. The equilibrium point $x^e$ is said to be globally asymptotically stable if for any initial state $x_0 \in \mathbb{R}^n$, $x_t \to x^e$ as $t \to \infty$. Now, we provide a vector comparison principle [35, 12, 26] for multi-dimensional ODE models, which plays a central role in the analysis below. We first introduce the quasi-monotone increasing function, which is a necessary prerequisite for the comparison principle.

**Definition 1** (Quasi-monotone function). *A vector-valued function $f : \mathbb{R}^n \to \mathbb{R}^n$ with $f := [f_1 \quad f_2 \quad \cdots \quad f_n]^T$ is said to be quasi-monotone increasing if $f_i(x) \le f_i(y)$ holds for all $i \in \{1, 2, \ldots, n\}$ and $x, y \in \mathbb{R}^n$ such that $x_i = y_i$ and $x_j \le y_j$ for all $j \ne i$.*

An example of a quasi-monotone increasing function $f$ is $f(x) = Ax$ where $A$ is a Metzler matrix, which implies the off-diagonal elements of $A$ are nonnegative. The vector comparison principle is presented below. For completeness, we provide a different proof tailored to our interests in the Appendix.

**Lemma 2** (Vector comparison principle [35, page 112], [12, Theorem 3.2]). *Suppose that $\overline{f}$ and $\underline{f}$ are globally Lipschitz continuous. Let $v_t$ be a solution of the system $\frac{d}{dt}x_t = \overline{f}(x_t), x_0 \in \mathbb{R}^n, \forall t \ge 0$, assume that $\overline{f}$ is quasi-monotone increasing, and let $v_t$ be a solution of the system*

$$\frac{d}{dt}v_t = \underline{f}(v_t), \quad v_0 < x_0, \quad \forall t \ge 0, \tag{2}$$

*where $\underline{f}(v) \le \overline{f}(v)$ holds for any $v \in \mathbb{R}^n$. Then, $v_t \le x_t$ for all $t \ge 0$.*

## 2.3 Switching system theory

Consider the particular nonlinear system, the *linear switching system*,

$$\frac{d}{dt}x_t = A_{\sigma_t}x_t, \quad x_0 = z \in \mathbb{R}^n, \quad t \in \mathbb{R}_+, \tag{3}$$

where $x_t \in \mathbb{R}^n$ is the state, $\sigma \in \mathcal{M} := \{1, 2, \ldots, M\}$ is called the mode, $\sigma_t \in \mathcal{M}$ is called the switching signal, and $\{A_\sigma, \sigma \in \mathcal{M}\}$ are called the subsystem matrices. The switching signal can be either arbitrary or controlled by the user under a certain switching policy. Especially, a state-feedback switching policy is denoted by $\sigma(x_t)$. The global asymptotic stability of the switching system is guaranteed under a fundamental algebraic stability condition reported in [22].

**Lemma 3** ([22, Theorem 8]). *The origin of the linear switching system* (3) *is the unique globally asymptotically stable equilibrium point under arbitrary switchings, $\sigma_t$, if and only if there exist a full column rank matrix, $L \in \mathbb{R}^{m \times n}$, $m \ge n$, and a family of matrices, $\bar{A}_\sigma \in \mathbb{R}^{m \times n}$, $\sigma \in \mathcal{M}$, with the so-called "strictly negative row dominating diagonal condition", i.e., for each $\bar{A}_\sigma, \sigma \in \mathcal{M}$, its elements satisfy*

$$[\bar{A}_\sigma]_{ii} + \sum_{j \in \{1,2,\ldots,n\}\backslash\{i\}} |[\bar{A}_\sigma]_{ij}| < 0, \quad \forall i \in \{1, 2, \ldots, m\},$$

*such that the following matrix relation is satisfied: $LA_\sigma = \bar{A}_\sigma L, \quad \forall \sigma \in \mathcal{M}$.*

## 2.4 ODE-based stochastic approximation

Because of its generality, the convergence analyses of many RL algorithms rely on the ODE approach [3, 17]. It analyzes convergence of general stochastic recursions by examining stability of the associated ODE model based on the fact that the stochastic recursions with diminishing step-sizes approximate the corresponding ODEs in the limit. One of the most popular approach is based on the Borkar and Meyn theorem [4]. We now briefly introduce the Borkar and Meyn's ODE approach for analyzing convergence of the general stochastic recursions

$$\theta_{k+1} = \theta_k + \alpha_k(f(\theta_k) + \varepsilon_{k+1}) \tag{4}$$

where $f : \mathbb{R}^n \to \mathbb{R}^n$ is a nonlinear mapping. Basic technical assumptions are given below.

**Assumption 1.**

1. *The mapping $f : \mathbb{R}^n \to \mathbb{R}^n$ is globally Lipschitz continuous and there exists a function $f_\infty : \mathbb{R}^n \to \mathbb{R}^n$ such that $\lim_{c \to \infty} \frac{f(cx)}{c} = f_\infty(x), \forall x \in \mathbb{R}^n$.*

2. *The origin in $\mathbb{R}^n$ is an asymptotically stable equilibrium for the ODE $\dot{x}_t = f_\infty(x_t)$.*

3. *There exists a unique globally asymptotically stable equilibrium $\theta^e \in \mathbb{R}^n$ for the ODE $\dot{x}_t = f(x_t)$, i.e., $x_t \to \theta^e$ as $t \to \infty$.*

4. *The sequence $\{\varepsilon_k, \mathcal{G}_k, k \geq 1\}$ with $\mathcal{G}_k = \sigma(\theta_i, \varepsilon_i, i \leq k)$ is a martingale difference sequence. In addition, there exists a constant $C_0 < \infty$ such that for any initial $\theta_0 \in \mathbb{R}^n$, we have $\mathbb{E}[\|\varepsilon_{k+1}\|^2 | \mathcal{G}_k] \leq C_0(1 + \|\theta_k\|^2), \forall k \geq 0$.*

5. *The step-sizes satisfy $\alpha_k > 0, \sum_{k=0}^{\infty} \alpha_k = \infty, \sum_{k=0}^{\infty} \alpha_k^2 < \infty$.*

**Lemma 4** ([4, Borkar and Meyn theorem]). *Under Assumption 1, for any initial $\theta_0 \in \mathbb{R}^n$, $\sup_{k \geq 0} \|\theta_k\| < \infty$ with probability one. In addition, $\theta_k \to \theta^e$ as $k \to \infty$ with probability one.*

## 3 Convergence Analysis of Asynchronous Q-learning

We consider the Q-learning updates

$$Q_{k+1}(s_k, a_k) = Q_k(s_k, a_k) + \alpha_k \left\{ r_{a_k}(s_k, s_{k+1}) + \gamma \max_{a \in \mathcal{A}} Q_k(s_{k+1}, a_k) - Q_k(s_k, a_k) \right\}, \tag{5}$$

where $\alpha_k \geq 0$ is the learning rate and $(s_k, a_k, s_{k+1})$ comes from the trajectory of some behavior policy. For simplicity of presentation, we assume $\{(s_k, a_k)\}_{k=0}^{\infty}$ is a sequence of i.i.d. random variables from the stationary state-action distribution, $d_a(s)$, such that $d_a(s) > 0$ holds for all $s \in \mathcal{S}, a \in \mathcal{A}$. This assumption is common in the ODE approaches for Q-learning and TD-learning [29] and can potentially be relaxed. Note that different from the original Watkin's Q-learning, we do not require the step-size $\alpha_k$ to depend on the state-action pair.

Before proceeding, we introduce the following compact notations:

$$P := [P_1^T, \cdots, P_{|\mathcal{A}|}^T]^T \in \mathbb{R}^{|\mathcal{S}| \times |\mathcal{S}||\mathcal{A}|}, \ R := [R_1^T, \cdots, R_{|\mathcal{A}|}^T]^T \in \mathbb{R}^{|\mathcal{S}||\mathcal{A}|},$$
$$D_a := \text{diag}[d_a(1), \cdots, d_a(|\mathcal{S}|)] \in \mathbb{R}^{|\mathcal{S}| \times |\mathcal{S}|}, \ D := \text{diag}[D_1, \cdots, D_{|\mathcal{A}|}] \in \mathbb{R}^{|\mathcal{S}||\mathcal{A}| \times |\mathcal{S}||\mathcal{A}|}.$$

and $Q := [Q_1^T, \cdots, Q_{|\mathcal{A}|}^T]^T \in \mathbb{R}^{|\mathcal{S}||\mathcal{A}|}$, where $Q_a = Q(\cdot, a) \in \mathbb{R}^{|\mathcal{S}|}$, and $R_a(s) := \mathbb{E}[r_a(s, s')|s, a]$. By definition, $D$ is a nonsingular diagonal matrix with strictly positive diagonal elements. In addition, we denote $e_s \in \mathbb{R}^{|\mathcal{S}|}$ and $e_a \in \mathbb{R}^{|\mathcal{A}|}$ as $s$-th basis vector (zero except for the $s$-th component) and $a$-th basis vector, respectively. For any deterministic policy, $\pi : \mathcal{S} \to \mathcal{A}$, we define the corresponding distribution vector $\vec{\pi}(s) := e_{\pi(s)} \in \Delta_{|\mathcal{S}|}$, where $\Delta_{|\mathcal{S}|}$ is the set of all probability distributions over $\mathcal{S}$. Lastly, we denote the matrix

$$\Pi_\pi := [\vec{\pi}(1) \otimes e_1, \vec{\pi}(2) \otimes e_2, \ldots, \vec{\pi}(|S|) \otimes e_{|S|}]^T \in \mathbb{R}^{|\mathcal{S}| \times |\mathcal{S}||\mathcal{A}|}$$

and greedy policy $\pi_Q(s) := \arg\max_{a \in \mathcal{A}} e_s^T Q_a \in \mathcal{A}$. By definition, for any $\pi \in \Theta$, $P\Pi_\pi$ is the state-action pair transition probability matrix under the deterministic policy $\pi$.

## 3.1 Asynchronous Q-learning as affine switching system

Using the notation introduced, the Q-learning update can be rewritten as

$$Q_{k+1} = Q_k + \alpha_k \left( DR + \gamma DP\Pi_{\pi_{Q_k}} Q_k - DQ_k + \varepsilon_{k+1} \right), \tag{6}$$

where

$$\varepsilon_{k+1} = (e_a \otimes e_s)(e_a \otimes e_s)^T R + \gamma (e_a \otimes e_s)(e_{s'})^T \Pi_{\pi_{Q_k}} Q_k$$
$$- (e_a \otimes e_s)(e_a \otimes e_s)^T Q_k - (DR + \gamma DP\Pi_{\pi_{Q_k}} Q_k - DQ_k).$$

It can be easily shown that $\{\varepsilon_{k+1}\}$ is a martingale difference sequence. Using the Bellman equation $(\gamma DP\Pi_{\pi_{Q^*}} - D)Q^* + DR = 0$, (6) can be further rewritten as

$$(Q_{k+1} - Q^*) = (Q_k - Q^*) + \alpha_k \big[ (\gamma DP\Pi_{\pi_{Q_k}} - D)(Q_k - Q^*)$$
$$+ \gamma DP(\Pi_{\pi_{Q_k}} - \Pi_{\pi_{Q^*}})Q^* + \varepsilon_{k+1} \big]. \tag{7}$$

As discussed in Section 2.4, the convergence of (7) can be analyzed by evaluating the stability of the corresponding continuous-time ODE

$$\frac{d}{dt}(Q_t - Q^*) = (\gamma DP\Pi_{\pi_{Q_t}} - D)(Q_t - Q^*) + \gamma DP(\Pi_{\pi_{Q_t}} - \Pi_{\pi_{Q^*}})Q^*, \, Q_0 - Q^* = z \in \mathbb{R}^{|\mathcal{S}||\mathcal{A}|}, \tag{8}$$

which is an affine switching system. More precisely, if we define a one-to-one map $\psi : \Theta \to \{1, 2, \ldots, |\Theta|\}$, where $\Theta$ is the set of all deterministic policies, $x_t := Q_t - Q^*$, and

$$(A_{\psi(\pi)}, b_{\psi(\pi)}) := (\gamma DP\Pi_\pi - D, \gamma DP(\Pi_\pi - \Pi_{\pi_{Q^*}})Q^*)$$

for all $\pi \in \Theta$, then (8) can be represented by the affine switching system

$$\frac{d}{dt}x_t = A_{\sigma(x_t)}x_t + b_{\sigma(x_t)}, \quad x_0 = z \in \mathbb{R}^{|\mathcal{S}||\mathcal{A}|}, \tag{9}$$

where, $\sigma : \mathbb{R}^{|\mathcal{S}||\mathcal{A}|} \to \{1, 2, \ldots, |\Theta|\}$ is a state-feedback switching policy defined by $\sigma(x_t) := \psi(\pi_{Q_t}), \pi_{Q_t}(s) = \arg\max_{a \in \mathcal{A}} e_s^T Q_{t,a}$.

Since (9) is a switching system with a state-feedback switching policy, it may cause arbitrary switching behaviors. It is unclear whether its solution exists over all $t \geq 0$ and whether the solution is unique. We establish the existence and uniqueness of its solution, which follows from the global Lipschitz continuity of the affine mapping.

**Proposition 1.** *The mapping $f(\theta) = (\gamma DP\Pi_{\pi_\theta} - D)\theta$. is globally Lipschitz continuous w.r.t. $\|\cdot\|_\infty$. Hence, the solution of the switching system (9) exists and is unique for all $t \geq 0$ and $x_0 \in \mathbb{R}^n$.*

## 3.2 Stability analysis

Note that proving the global asymptotic stability of (9) without the affine term is relatively straightforward based on Lemma 3. However, none of the existing theory supports switching systems with affine terms. To address this issue, we construct two comparison systems by exploiting the special structure of the switching system and the greedy policy and prove their global asymptotic stability. By further building on the vector comparison principle introduced in Lemma 2, we then establish the asymptotic stability of the desired affine switching system.

More specifically, we consider the upper comparison system

$$\frac{d}{dt}(Q_t^u - Q^*) = (\gamma DP\Pi_{\pi_{Q_t^u}} - D)(Q_t^u - Q^*), \quad Q_0^u - Q^* > Q_0 - Q^* \in \mathbb{R}^{|\mathcal{S}||\mathcal{A}|}, \tag{10}$$

and the lower comparison system

$$\frac{d}{dt}(Q_t^l - Q^*) = (\gamma DP\Pi_{Q^*} - D)(Q_t^l - Q^*), \quad Q_0^l - Q^* < Q_0 - Q^* \in \mathbb{R}^{|\mathcal{S}||\mathcal{A}|}. \tag{11}$$

Observe that, (10) is a linear switching system and (11) is a linear system. We can prove that both systems are asymptotically stable by verifying the strictly negative row dominating diagonal condition required in Lemma 3. By using the vector comparison theorem and the quasi-monotone property, we can prove that the original switching affine system's trajectories are sandwiched by the trajectories of the two systems.

**Theorem 1.** *Consider the systems (8), (10) and(11). We have*

1. $Q_t^l - Q^* \leq Q_t - Q^* \leq Q_t^u - Q^*$ for all $t \geq 0$;

2. The origin is the unique globally asymptotically stable equilibrium point of the affine switching system (8).

We are now in position to apply the Borkar and Meyn theorem to establishing the convergence of asynchronous Q-learning.

**Theorem 2.** *Assume that the step-sizes satisfy*

$$\alpha_k > 0, \quad \sum_{k=0}^{\infty} \alpha_k = \infty, \quad \sum_{k=0}^{\infty} \alpha_k^2 < \infty. \tag{12}$$

*Then, $Q_k \to Q^*$ with probability one.*

The proof is fairly straightforward by invoking Lemma 4, Proposition 1, and Theorem 1. We can see that the convergence of asynchronous Q-learning follows immediately after proving the asymptotic stability of the associated affine switching system. In contrast, the convergence analysis of Q-learning in [4] relies on a nonlinear ODE model, whose asymptotic stability is proved in [5] by using the max-norm contraction of the Bellman operator; yet the analysis only applies to *synchronous Q-learning*, i.e., at each time all entries of the iterate are updated.

Lastly, it is worth mentioning that a number of work has recently established non-asymptotic analysis for asynchronous Q-learning, including [32, 9, 27]. The current best known bound is given in [27] showing a complexity of $\mathcal{O}\left(\frac{(|\mathcal{S}||\mathcal{A}|)^2}{(1-\gamma)^5 \epsilon^2}\right)$. However, we stress that unlike this line of work, the purpose of our work is not to provide a tight convergence rate for asynchronous Q-learning, but rather to build an intuitive understanding of the family of Q-learning algorithms through the lens of switching systems. The switching system framework provides a simpler analysis and can be easily extended to deal with many Q-learning variants, as we show in the subsequent sections.

## 4  Convergence Analysis of Averaging Q-learning

We now consider a variant of the asynchronous Q-learning algorithm, called *averaging Q-learning*, which is newly introduced in [19] and motivated by the success of deep Q-learning [25], in order to improve the stability. The averaging Q-learning maintains two separate estimates, the target estimate and the online estimate: the online estimate is for approximating the state-action value function $Q$ and updated through an online manner, whereas the target estimate is for computing the target values and updated through taking Polyak's averaging. Specifically, the algorithm works as follows:

$$Q_{k+1}^A(s_k, a_k) = Q_k^A(s_k, a_k) + \alpha_k \left\{ r_{a_k}(s_k, s_{k+1}) + \gamma \max_{a \in \mathcal{A}} Q_k^B(s_{k+1}, a_k) - Q_k^A(s_k, a_k) \right\}, \tag{13}$$

$$Q_{k+1}^B(s_k, a_k) = Q_k^B(s_k, a_k) + \alpha_k \delta(Q_k^A(s_k, a_k) - Q_k^B(s_k, a_k)). \tag{14}$$

where $\delta > 0$ is a rescaling constant. Following similar arguments as in the asynchronous Q-learning case, the corresponding ODE model is given by the following switching system:

$$\frac{d}{dt} \begin{bmatrix} Q_t^A - Q^* \\ Q_t^B - Q^* \end{bmatrix} = \begin{bmatrix} -D & \gamma DP\Pi_{\pi_{Q_t^B}} \\ \delta I & -\delta I \end{bmatrix} \begin{bmatrix} Q_t^A - Q^* \\ Q_t^B - Q^* \end{bmatrix} + \begin{bmatrix} \gamma DP(\Pi_{\pi_{Q_t^B}} - \Pi_{\pi_{Q^*}})Q^* \\ 0 \end{bmatrix},$$

$$\begin{bmatrix} Q_0^A - Q^* \\ Q_0^B - Q^* \end{bmatrix} = z \in \mathbb{R}^{2|\mathcal{S}||\mathcal{A}|}, \tag{15}$$

which matches with the general form in (9). We obtain the global asymptotic stability of (15).

**Theorem 3.** *For any $\delta > 0$, the origin is the unique globally asymptotically stable equilibrium point of the affine switching system* (15).

As a result, by invoking Borkar and Meyn's theorem similarly as before, we arrive at

**Theorem 4.** *For the averaging Q-learning, assuming the step-sizes satisfy* (12)*, then for any $\delta > 0$, $Q_k^A \to Q^*$ and $Q_k^B \to Q^*$ with probability one.*

We remark that this is indeed the first convergence analysis of the averaging Q-learning algorithm. In contrast, previous work [19] only provided the asymptotic convergence of averaging TD-learning. We expect that this analysis would also shed light on the convergence of other target-based Q-learning algorithms, e.g., the double Q-learning [11], periodic Q-learning [20], etc.

# 5 Convergence Analysis of Q-learning with Linear Function Approximation

When the state-space is large, linear function approximation are commonly used to approximate the optimal Q-function, $Q^* \cong \Phi\theta^*$, where $\Phi$ is the feature matrix. In particular, given pre-selected basis (or feature) functions $\phi_1, \ldots, \phi_n : \mathcal{S} \to \mathbb{R}$, the feature matrix $\Phi \in \mathbb{R}^{|\mathcal{S}| \times n}$ is defined as $\Phi := [\phi(1,1), \phi(2,1), \cdots, \phi(|\mathcal{S}|, |\mathcal{A}|)]^T \in \mathbb{R}^{|\mathcal{S}||\mathcal{A}| \times n}$, where $\phi(s,a)^T := [\phi_1(s,a), \phi_2(s,a), \cdots, \phi_n(s,a)] \in \mathbb{R}^n$. Here $n \ll |\mathcal{S}||\mathcal{A}|$ is a positive integer.

Q-learning with linear function approximation performs the following update:

$$\theta_{k+1} = \theta_k + \alpha_k \phi(s_k, a_k)\big[r_a(s_k, s_{k+1}) + \gamma \max_{a \in \mathcal{A}}(\Phi\theta_k)(s_{k+1}, a) - (\Phi\theta_k)(s_k, a_k)\big], \qquad (16)$$

where $\alpha_k \geq 0$ is the learning rate and $\{(s_k, a_k)\}_{k=0}^{\infty}$ are sampled from the stationary state-action distribution $d_a(s)$ under a behavior policy $\beta$ such that $d_a(s) = \lim_{k \to \infty} \mathbb{P}[(s_k, a_k) = (s, a)|\beta]$. It is well-known that Q-learning with linear function approximation may not converge in general [30]. Under certain conditions, its convergence can be proven. For instance, [24] demonstrates the asymptotic convergence assuming that the following condition holds:

$$\gamma^2 \Phi^T \Pi_\pi^T D^\beta \Pi_\pi \Phi \prec \Phi^T D\Phi, \quad \forall \pi \in \Theta_\Phi, \qquad (17)$$

where $\Theta_\Phi := \{\pi \in \Theta : \pi(s) = \arg\max_{a \in \mathcal{A}}(\Phi\theta)(s,a), \forall s \in \mathcal{S}, \theta \in \mathbb{R}^m\}$ and $D^\beta$ is a diagonal matrix whose diagonal entries correspond to the stationary state distribution of the underlying Markov decision process under the behavior policy $\beta$. Recently, [6] considered a slightly stronger condition in order to obtain the convergence rate of Q-learning with linear function approximation.

In this section, we analyze the convergence from the switching system perspective and provide a new sufficient condition that ensures the asymptotic convergence. We start by introducing some basic assumptions.

**Assumption 2.** $[\Phi]_{ij} \geq 0$ for all $i \in \mathcal{S}$ and $j \in \{1, 2, \ldots, n\}$.

**Assumption 3.** All column vectors of $\Phi$ are orthogonal.

Assumption 2 requires all elements of $\Phi$ to be nonnegative. This assumption is required in our convergence analysis to obtain lower and upper comparison systems of the affine switching system. In the case that no function approximation is used, $\Phi$ is set to be an identity matrix, $\Phi = I$, which automatically satisfies Assumption 2. We emphasize that this assumption is not very restrictive. For instance, if the values of rewards are nonnegative, then it is sufficient to set feature vectors with nonnegative elements when approximating the Q-function. Otherwise, the rewards can always be shifted to nonnegative by adding a large enough constant. Assumption 3 is slightly stricter than the assumption of having full column rank which is usually adopted in the RL literature. This is required in order to guarantee the quasi-monotonicity of the corresponding switching system models.

Following a similar analysis, the associated affine switching system is given by

$$\frac{d}{dt}\theta_t = (\gamma\Phi^T DP\Pi_{\pi_{\Phi\theta_t}}\Phi - \Phi^T D\Phi)\theta_t + \Phi^T DR, \quad \theta_0 \in \mathbb{R}^n,$$

or equivalently,

$$\frac{d}{dt}(\theta_t - \theta^*) = (\gamma\Phi^T DP\Pi_{\pi_{\Phi\theta_t}}\Phi - \Phi^T D\Phi)(\theta_t - \theta^*) + \gamma\Phi^T DP(\Pi_{\pi_{\Phi\theta_t}} - \Pi_{\pi_{\Phi\theta^*}})\Phi\theta^*, \quad (18)$$

where $\theta_0 - \theta^* = z \in \mathbb{R}^n$, $\pi_{\Phi\theta_t}(s) = \arg\max_{a \in \mathcal{A}}(\Phi\theta_t)(s,a)$ and $\theta^*$ is the optimal parameter satisfying the projected Bellman equation $\Phi\theta^* = \Gamma(\gamma P\Pi_{\pi_{\Phi\theta^*}}\Phi\theta^* + R)$, and $\Gamma := \Phi(\Phi^T D\Phi)^{-1}\Phi^T D$ is the projection onto the range of $\Phi$.

We first establish the asymptotic stability of the system (18).

**Theorem 5.** *Suppose that Assumption 2 and Assumption 3 hold. The origin is the unique globally asymptotically stable equilibrium point of the affine switching system* (18) *if the following condition holds:*

$$-\phi_i^T D\phi_i + \phi_i^T \gamma DP\Pi_{\psi(\pi)} \sum_{j \in \{1,2,\ldots,n\}} \phi_j < 0, \quad \pi \in \Theta_\Phi, \qquad (19)$$

*where $\phi_i^T$ is the $i$-th row of the feature matrix $\Phi$.*

As a result, this leads to the following convergence:

**Theorem 6.** *For Q-learning with linear function approximation, under Assumptions 2-3 and the condition specified in (19), we have $\theta_k \to \theta^*$ with probability one.*

We now make some remarks on the sufficient condition (19), which may look abstract at first sight since it purely stems from switching system theory. Similar to Melo's condition, our new condition also suggests that the behavior policy should be close to the optimal policy. In fact, this condition is quite similar to the diagonal dominant condition used in network science fields [37, 7]. Our analysis indicates that this condition is a necessary and sufficient condition for the asymptotic stability of the underlying switching system model of Q-learning, while Melo's condition is only a sufficient condition for the asymptotic stability. Especially, Melo's condition is strong enough to guarantee the existence of a quadratic Lyapunov function for the underlying switching system model, while in general, the switching system does not necessarily admit quadratic Lyapunov functions. This shows the less conservativeness of our new condition.

**Proposition 2.** *Under the above assumptions, Melo's condition* (17) *implies the condition (19).*

In practice, to derive a computationally tractable sufficient condition, $\Theta_\Phi$ can be replaced with $\Theta$. A special case where the condition (19) holds is when elements of the feature vectors $\phi_i$ are binary numbers, $\{0, 1\}$. This clearly holds for the tabular setting.

**Proposition 3.** *Suppose the elements of the feature matrix $\Phi$ are binary numbers, i.e., $\{0, 1\}$, then the condition (19) always holds.*

Lastly, we give a simple MDP example which satisfies the sufficient condition in (19), but violates the Melo's condition (17).

**Example 1.** *Consider an MDP with $\mathcal{S} = \{1, 2\}$, $\mathcal{A} = \{1, 2\}$, $\gamma = 0.9$, $P_1 = \begin{bmatrix} 1/2 & 1/2 \\ 1 & 0 \end{bmatrix}$, $P_2 = \begin{bmatrix} 0 & 1 \\ 2/3 & 1/3 \end{bmatrix}$, and a behavior policy $\beta$ such that $\mathbb{P}[a = 1|s = 1] = 0.2, \mathbb{P}[a = 2|s = 1] = 0.8, \mathbb{P}[a = 1|s = 2] = 0.7, \quad \mathbb{P}[a = 2|s = 2] = 0.3$. The corresponding matrices $D^\beta$ and $D$ are given by*

$$D^\beta = \begin{bmatrix} 0.5 & 0 \\ 0 & 0.5 \end{bmatrix}, \quad D = \begin{bmatrix} 0.1 & 0 & 0 & 0 \\ 0 & 0.35 & 0 & 0 \\ 0 & 0 & 0.4 & 0 \\ 0 & 0 & 0 & 0.15 \end{bmatrix}.$$

*If the feature matrix is $\Phi^T = \begin{bmatrix} 1 & 2 & 0 & 1 \end{bmatrix}$, then $\Theta_\Phi$ is given by $\Theta_\Phi = \{\pi_1, \pi_2\}$, where $\pi_1$ is a deterministic policy such that $\pi_1(1) = 1$, $\pi_1(2) = 1$ and $\pi_2$ such that $\pi_2(1) = 2$, $\pi_2(2) = 2$, which are obtained by considering three cases, $\theta > 0, \theta = 0, \theta < 0$, Here, we assume that whenever $\{1, 2\} = \arg\max_{a \in \mathcal{A}}(\Phi\theta)(s, a)$, we select $a = 1$ in Q-learning. The quantities in (19) are given by $-0.885$ and $-0.03$ for all $\pi \in \Theta_\Phi = \{\pi_1, \pi_2\}$, implying convergence of the algorithm. However, the quantity $\gamma^2 \Phi^T \Pi_\pi^T D^\beta \Pi_\pi \Phi - \Phi^T D \Phi$ is computed as $-1.2450$ and $0.3750$ for all $\pi \in \Theta_\Phi = \{\pi_1, \pi_2\}$, respectively. This implies that the condition in (17) fails to verify the convergence.*

## 6 Numerical Simulation

Consider an MDP with $\mathcal{S} = \{1, 2\}$, $\mathcal{A} = \{1, 2\}$, $\gamma = 0.9$,

$$P_1 = \begin{bmatrix} 0.2 & 0.8 \\ 0.3 & 0.7 \end{bmatrix}, \quad P_2 = \begin{bmatrix} 0.5 & 0.5 \\ 0.7 & 0.3 \end{bmatrix}, R_1 = \begin{bmatrix} 3 \\ 1 \end{bmatrix}, \quad R_2 = \begin{bmatrix} 2 \\ 1 \end{bmatrix}$$

and a behavior policy $\beta$ such that

$$\mathbb{P}[a = 1|s = 1] = 0.2, \quad \mathbb{P}[a = 2|s = 1] = 0.8,$$
$$\mathbb{P}[a = 1|s = 2] = 0.7, \quad \mathbb{P}[a = 2|s = 2] = 0.3.$$

Simulated trajectories of the O.D.E. model of Q-learning including the upper and lower comparison systems are depicted in Figure 1. Simulated trajectories of the O.D.E. model of the averaging Q-learning including the upper and lower comparison systems are depicted in Figure 2 for $Q_t^A$ part. The simulation study empirically justifies the bounding principles and asymptotic convergence established in theory.

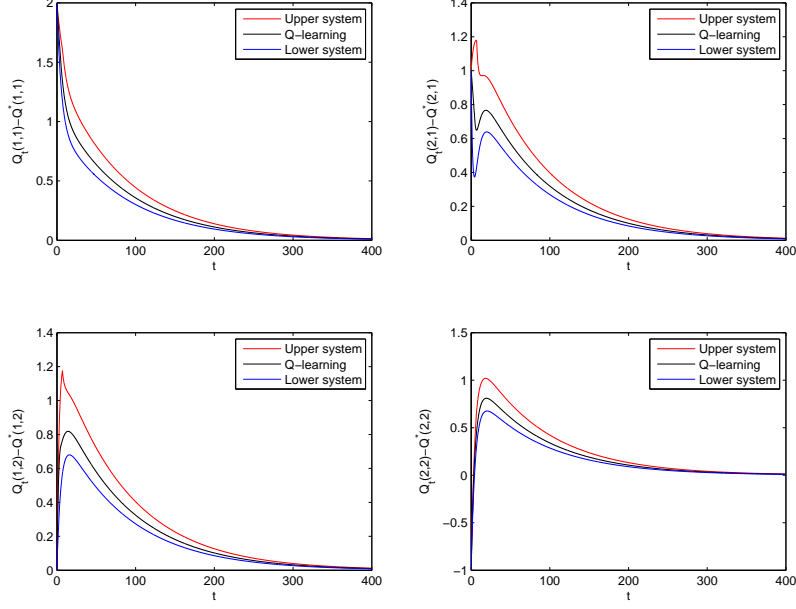

Figure 1: Trajectories of the O.D.E. model of Q-learning and the upper and lower comparison systems

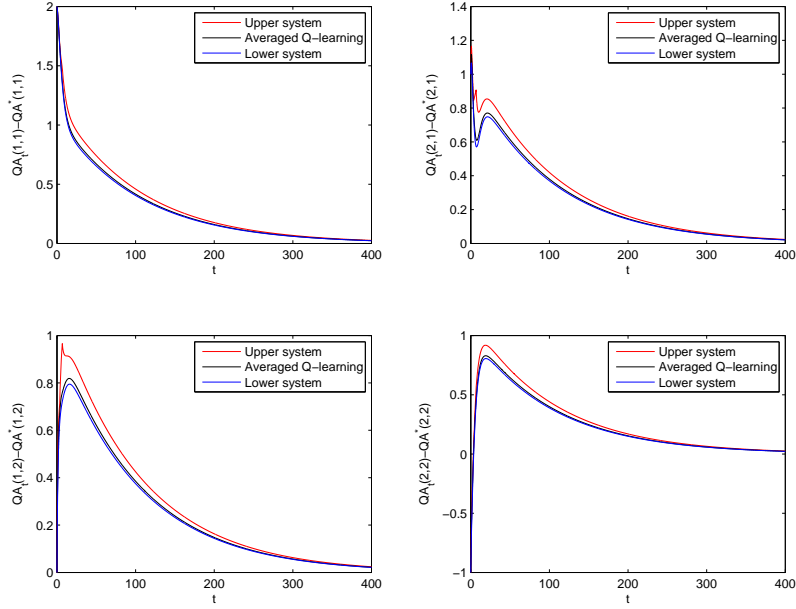

Figure 2: Trajectories of the O.D.E. model of averaging Q-learning and the upper and lower comparison systems ($Q_t^A$ part)

## 7 Conclusion

In this paper, we offer a unified and convenient convergence analysis of various Q-learning algorithms building on novel connections to switching systems. We establish the first ODE analysis for asynchronous Q-learning and averaging Q-learning, and derive a new sufficient condition to ensure the convergence of Q-learning with linear function approximation. While this work focuses only on the asymptotic convergence of a subset of RL algorithms, we expect that the switching system approach could be leveraged to advance the understanding of many other RL algorithms and extended to non-asymptotic analysis. This may also shed light on the design of more efficient and robust RL algorithms from the control perspective, which we leave for future investigation.

## Acknowledgments and Disclosure of Funding

We thank the reviewers and area chair for constructive feedback. We would like to thank Csaba Szepesvari, Bin Hu, and Rohit Gupta for insightful comments. The work was supported by NSF CRII 1755829 and NSF CCF 1934986.

## Broader Impact

By bridging Q-learning with switching systems, this work has full potential to promote synergy between two closely related fields/communities: control theory and reinforcement learning, as well as to stimulate further developments in the theory, algorithms and applicability of reinforcement learning. Meanwhile, this paper provides an accessible material on the basics of stochastic approximation, switching system theory, and reinforcement learning theory, which could be beneficial to graduate students, researchers, and even reinforcement learning practitioners. Our analysis could potentially inspire the development of more efficient and robust algorithms that benefit a broad spectrum of data-intensive applications in the realm of reinforcement learning.

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
