[Supplementary Material]

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

6. $P_a(s, s')$: the state transition probability from the current state $s \in \mathcal{S}$ to the next state $s' \in \mathcal{S}$ under action $a \in \mathcal{A}$
7. $r_a(s, s')$: the reward random variable conditioned on $a \in \mathcal{A}, s, s' \in \mathcal{S}$
8. $R_a(s, s') := \mathbb{E}[r_a(s, s')|s, a, s']$
9.
$$
P := \begin{bmatrix} P_1 \\ \vdots \\ P_{|\mathcal{A}|} \end{bmatrix} \in \mathbb{R}^{|\mathcal{S}| \times |\mathcal{S}||\mathcal{A}|}, \quad R := \begin{bmatrix} R_1 \\ \vdots \\ R_{|\mathcal{A}|} \end{bmatrix} \in \mathbb{R}^{|\mathcal{S}||\mathcal{A}|}, \quad Q := \begin{bmatrix} Q_1 \\ \vdots \\ Q_{|\mathcal{A}|} \end{bmatrix} \in \mathbb{R}^{|\mathcal{S}||\mathcal{A}|}
$$
where $Q_a = Q(\cdot, a) \in \mathbb{R}^{|\mathcal{S}|}, a \in \mathcal{A}$ and $R_a(s) := \mathbb{E}[r_a(s, s')|s, a]$.
10.
$$
D_a := \begin{bmatrix} d_a(1) & & \\ & \ddots & \\ & & d_a(|\mathcal{S}|) \end{bmatrix} \in \mathbb{R}^{|\mathcal{S}| \times |\mathcal{S}|}, \quad D := \begin{bmatrix} D_1 & & \\ & \ddots & \\ & & D_{|\mathcal{A}|} \end{bmatrix} \in \mathbb{R}^{|\mathcal{S}||\mathcal{A}| \times |\mathcal{S}||\mathcal{A}|}
$$
11. $e_s \in \mathbb{R}^{|\mathcal{S}|}$ and $e_a \in \mathbb{R}^{|\mathcal{A}|}$: $s$-th basis vector (zero except for the $s$-th component) and $a$-th basis vector, respectively
12. $\Delta_{|\mathcal{S}|}$: the set of all probability distributions over $\mathcal{S}$
13. $\vec{\pi}(s) := e_{\pi(s)} \in \Delta_{|\mathcal{S}|}$
14.
$$
\Pi_\pi := \begin{bmatrix} \vec{\pi}(1)^T \otimes e_1^T \\ \vec{\pi}(2)^T \otimes e_2^T \\ \vdots \\ \vec{\pi}(|S|)^T \otimes e_{|S|}^T \end{bmatrix} \in \mathbb{R}^{|\mathcal{S}| \times |\mathcal{S}||\mathcal{A}|}.
$$
15. Feature vector: $\phi(s, a)^T := [\phi_1(s, a), \phi_2(s, a), \cdots, \phi_n(s, a)] \in \mathbb{R}^n$.
16. Feature matrix:
$$
\Phi := \begin{bmatrix} \phi(1, 1)^T \\ \phi(2, 1)^T \\ \vdots \\ \phi(|\mathcal{S}|, |\mathcal{A}|)^T \end{bmatrix} \in \mathbb{R}^{|\mathcal{S}||\mathcal{A}| \times n},
$$

## B  Proof of Lemma 2

*Proof.* We simplify and summarize the ideas of the proofs in the literature, [35, page 112],[12, Theorem 3.2.], in the following proof. Instead of (2), first consider

$$
\frac{d}{dt} v_\varepsilon(t) = \underline{f}(v_\varepsilon(t)) - \varepsilon \mathbf{1}_n, \quad v_\varepsilon(0) < x(0), \quad \forall t \geq 0
$$

where $\varepsilon > 0$ is a sufficiently small real number and $\mathbf{1}_n$ is a vector where all elements are ones, where we use a different notation for the time index for convenience. Suppose that the statement is not true, and let

$$
t^* := \inf\{t \geq 0 : \exists i \text{ such that } v_{\varepsilon,i}(t) > x_i(t)\} < \infty,
$$

and let $i$ be such index. By the definition of $t^*$, we have that $v_{\varepsilon,i}(t^*) = x_i(t^*)$ and $v_{\varepsilon,j}(t^*) \le x_j(t^*)$ for any $j \ne i$. Then, since $\overline{f}$ is quasi-monotone increasing, we have

$$\overline{f}_i(v_\varepsilon(t^*)) \le \overline{f}_i(x(t^*)). \tag{20}$$

On the other hand, by the definition of $t^*$, there exists a small $\delta > 0$ such that

$$v_{\varepsilon,i}(t^* + \Delta t) > x_i(t^* + \Delta t)$$

for all $0 < \Delta t < \delta$. Dividing both sides by $\Delta t$ and taking the limit $\Delta t \to 0$, we have

$$\dot{v}_{\varepsilon,i}(t^*) \ge \dot{x}_i(t^*) = \overline{f}_i(x(t^*)). \tag{21}$$

By the hypothesis, it holds that

$$\frac{d}{dt} v_\varepsilon(t) = \underline{f}(v_\varepsilon(t)) - \varepsilon \mathbf{1}_n < \underline{f}(v_\varepsilon(t)) \le \overline{f}(v_\varepsilon(t))$$

holds for all $t \ge 0$. The inequality implies $\dot{v}_{\varepsilon,i}(t) < \overline{f}_i(v_\varepsilon(t))$, which in combination with (21) leads to $\overline{f}_i(v_\varepsilon(t^*)) > \overline{f}_i(x(t^*))$. However, it contradicts with (20). Therefore, $v_\varepsilon(t) \le x(t)$ holds for all $t \ge 0$. Since the solution $v_\varepsilon(t)$ continuously depends on $\varepsilon > 0$ [35, Chap. 13], taking the limit $\varepsilon \to 0$, we conclude $v_0(t) \le x(t)$ holds for all $t \ge 0$. This completes the proof.

$\square$

## C  Proof of Proposition 1

*Proof.* The proof is completed by the inequalities

$$\begin{aligned}
\|f(x) - f(y)\|_\infty =& \|(\gamma DP\Pi_{\pi_x} - D)x - (\gamma DP\Pi_{\pi_y} - D)y\|_\infty \\
\le& \|\gamma DP\|_\infty \|\Pi_{\pi_x}x - \Pi_{\pi_y}y\|_\infty + \|D\|_\infty \|x - y\|_\infty \\
=& \|\gamma DP\|_\infty \max_{s \in \mathcal{S}} |\max_{a \in \mathcal{A}} x_a(s) - \max_{a \in \mathcal{A}} y_a(s)| + \|D\|_\infty \|x - y\|_\infty \\
\le& \|\gamma DP\|_\infty \max_{s \in \mathcal{S}} \max_{a \in \mathcal{A}} |x_a(s) - y_a(s)| + \|D\|_\infty \|x - y\|_\infty \\
=& \|\gamma DP\|_\infty \|x - y\|_\infty + \|D\|_\infty \|x - y\|_\infty \\
\le& (\|\gamma DP\|_\infty + \|D\|_\infty) \|x - y\|_\infty,
\end{aligned}$$

indicating that $f$ is globally Lipschitz continuous with respect to the $\|\cdot\|_\infty$ norm. $\square$

## D  Proof of Theorem 1

**Lemma 5.** *Consider the affine switching system* (9). *The origin of the associated linear switching system*

$$\frac{d}{dt} x_t = A_{\sigma_t} x_t,$$

*is the unique globally asymptotically stable equilibrium point under arbitrary switchings, $\sigma_t$.*

*Proof.* The proof follows by applying Lemma 3 with $L = I$, $\bar{A}_\sigma = A_\sigma$. In this case, the condition, $LA_\sigma = \bar{A}_\sigma L$ holds. It remains to prove the strictly negative row dominating diagonal property. For notational convenience, we definte $\Pi_\sigma$, $\sigma \in \mathcal{M}$ as $\Pi_{\pi_{Q_t^B}}$ such that $\sigma = \psi(\pi_{Q_t^B})$. Then,

$$\begin{aligned}
[A_\sigma]_{ii} + \sum_{j \in \{1,2,...,n\}\backslash\{i\}} |[A_\sigma]_{ij}| =& [D]_{ii}[\gamma P\Pi_\sigma - I]_{ii} + \sum_{j \in \{1,2,...,n\}\backslash\{i\}} [D]_{ii}|[\gamma P\Pi_\sigma - I]_{ij}| \\
\le& [\gamma P\Pi_\sigma - I]_{ii} + \sum_{j \in \{1,2,...,n\}\backslash\{i\}} |[\gamma P\Pi_\sigma - I]_{ij}| \\
=& [\gamma P\Pi_\sigma]_{ii} - 1 + \sum_{j \in \{1,2,...,n\}\backslash\{i\}} |[\gamma P\Pi_\sigma]_{ij}| \\
=& \gamma - 1 < 0, \quad \forall \sigma \in \mathcal{M},
\end{aligned}$$

which proves the global asymptotic stability. $\square$

*Proof of Theorem 1.* The basic idea of the proof is to find systems whose trajectories lower and upper bounds the trajectory of (9) by the vector comparison principle. Then, by proving the asymptotic stability of the two comparison systems, we can prove the asymptotic stability of (9).

Since each element of $\Pi_{\pi_{Q^*}}Q^*$ takes the maximum value across $a$, it is clear that $(\Pi_{\pi_{Q_t}} - \Pi_{\pi_{Q^*}})Q^* \leq 0$ holds, where the inequality is element-wise. Moreover, since $\gamma DP$ has nonnegative elements, $\gamma DP(\Pi_{\pi_{Q_t}} - \Pi_{\pi_{Q^*}})Q^* \leq 0$ holds. Therefore, we have $(\gamma D_\beta P\Pi_{\pi_{Q_t}} - D)(Q_t - Q^*) + \gamma DP(\Pi_{\pi_{Q_t}} - \Pi_{\pi_{Q^*}})Q^* \leq (\gamma DP\Pi_{\pi_{Q_t}} - D)(Q_t - Q^*) \leq (\gamma DP\Pi_{\pi_{Q_t - Q^*}} - D)(Q_t - Q^*)$ for all $t \in \mathbb{R}_+$. To proceed, define the vector functions

$$\overline{f}(y) = (\gamma DP\Pi_{\pi_y} - D)y,$$
$$\underline{f}(z) = (\gamma DP\Pi_{\pi_{z+Q^*}} - D)z + \gamma DP(\Pi_{\pi_{z+Q^*}} - \Pi_{\pi_{Q^*}})Q^*,$$

and consider the systems

$$\frac{d}{dt}y_t = \overline{f}(y_t), \quad y_0 > Q_0 - Q^*,$$
$$\frac{d}{dt}z_t = \underline{f}(z_t), \quad z_0 = Q_0 - Q^*,$$

for all $t \geq 0$. To apply Lemma 2, we will prove that $\overline{f}$ is quasi-monotone increasing. For any $z \in \mathbb{R}^{|\mathcal{S}||\mathcal{A}|}$, consider a nonnegative vector $p \in \mathbb{R}^{|\mathcal{S}||\mathcal{A}|}$ such that its $i$th element is zero. Then, for any $i \in \mathcal{S}$, we have

$$
\begin{aligned}
e_i^T \overline{f}(z+p) &= e_i^T(\gamma DP\Pi_{z+p} - D)(z+p) \\
&= \gamma e_i^T DP\Pi_{z+p}(z+p) - e_i^T Dz - e_i^T Dp \\
&= \gamma e_i^T DP\Pi_{z+p}(z+p) - e_i^T Dz \\
&= \gamma e_i^T DP \begin{bmatrix} \max_a(z_a(1) + p_a(1)) \\ \max_a(z_a(2) + p_a(2)) \\ \vdots \\ \max_a(z_a(|\mathcal{S}|) + p_a(|\mathcal{S}|)) \end{bmatrix} - e_i^T Dz \\
&\geq \gamma e_i^T DP \begin{bmatrix} \max_a z_a(1) \\ \max_a z_a(2) \\ \vdots \\ \max_a z_a(|S|) \end{bmatrix} - e_i^T Dz \\
&= e_i^T \overline{f}(z),
\end{aligned}
$$

which proves the quasi-monotone increasing property, where the second line is due to $e_i^T Dp = 0$. Moreover, following similar lines of the proof of Proposition 1, one can prove that $\overline{f}$ is Lipschitz continuous. Using $\underline{f}(z) = (\gamma DP\Pi_{\pi_{(z+Q^*)}} - D)(z + Q^*) + DR$ and following similar lines of the proof of Proposition 1, we conclude that $\underline{f}$ is Lipschitz continuous as well. Now, by Lemma 2, $Q_t - Q^* \leq Q_t^u - Q^*$ holds for every $t \in \mathbb{R}_+$, where $Q_t^u - Q^*$ is the solution of the switching system, which we refer to as an upper comparison system

$$\frac{d}{dt}(Q_t^u - Q^*) = (\gamma DP\Pi_{\pi_{Q_t^u}} - D)(Q_t^u - Q^*), \quad Q_0^u - Q^* > Q_0 - Q^* \in \mathbb{R}^{|\mathcal{S}||\mathcal{A}|},$$

By Lemma 5, the origin of the above switching system is globally asymptotically stable even under arbitrary switchings. Therefore, $Q_t - Q^*$ is asymptotically upper bounded by the zero vector as $t \to \infty$.

On the other hand, we have

$$
\begin{aligned}
(\gamma DP\Pi_{\pi_{Q_t}} - D)(Q_t - Q^*) + \gamma DP(\Pi_{\pi_{Q_t}} - \Pi_{\pi_{Q^*}})Q^* &= (\gamma DP\Pi_{\pi_{Q_t}} - D)Q_t + DR \\
&\geq (\gamma DP\Pi_{\pi_{Q^*}} - D)Q_t + DR = (\gamma DP\Pi_{\pi_{Q^*}} - D)(Q_t - Q^*),
\end{aligned}
$$

where the first inequality is due to $\gamma DP\Pi_{\pi_{Q_t}}Q_t \geq \gamma DP\Pi_{\pi_{Q^*}}Q_t$, and the second equality uses $DQ^* = \gamma DP\Pi_{\pi_{Q^*}}Q^* + DR$. Again, define the vector functions for lower comparison parts

$$\overline{f}(y) = (\gamma DP\Pi_{\pi_y} - D)y + DR,$$
$$\underline{f}(z) = (\gamma DP\Pi_{\pi_{Q^*}} - D)z + DR$$

and consider the systems

$$\frac{d}{dt}y_t = \overline{f}(y_t), \quad y_0 = Q_0,$$

$$\frac{d}{dt}z_t = \underline{f}(z_t), \quad z_0 < Q_0,$$

for all $t \geq 0$. To apply Lemma 2, we can prove that $\overline{f}$ is quasi-monotone increasing following the same lines as above. $\overline{f}$ is Lipschitz continuous by Proposition 1 and $\underline{f}$ is Lipschitz continuous as it is linear. Therefore, we can invoke Lemma 2, to prove the inequality $Q_t^{\overline{l}} - Q^* \leq Q_t - Q^*$ for all $t \geq 0$, where $Q_t^l - Q^*$ is the solution of the following linear system called the lower comparison system:

$$\frac{d}{dt}(Q_t^l - Q^*) = (\gamma DP\Pi_{Q^*} - D)(Q_t^l - Q^*), \quad Q_0^l - Q^* < Q_0 - Q^* \in \mathbb{R}^{|\mathcal{S}||\mathcal{A}|},$$

The origin of the above linear system is globally asymptotically stable equilibrium point by Lemma 5. Therefore, $Q_t - Q^*$ is asymptotically lower bounded by the zero vector as $t \to \infty$. Combining the bounds, we conclude that $Q_t - Q^* \to 0$ as $t \to \infty$. This completes the proof of Theorem 1. □

# E   Proof of Theorem 2

*Proof of Theorem 2.* First of all, note that the affine switching system model in (9) corresponds to the ODE model, $\frac{d}{dt}x_t = f(x_t)$, that appears in Assumption 1. The proof is completed by examining all the statements in Assumption 1:

1. Q-learning in (7) can be expressed as the stochastic recursion in (4) with
$$f(\theta) = (\gamma DP\Pi_{\pi_\theta} - D)\theta + \gamma DP(\Pi_{\pi_\theta} - \Pi_{\pi_{Q^*}})Q^*.$$
   To prove the first statement of Assumption 1, we note that
$$\frac{f(c\theta)}{c} = (\gamma DP\Pi_{\pi_\theta} - D)\theta + \frac{\gamma DP(\Pi_{\pi_\theta} - \Pi_{\pi_{Q^*}})Q^*}{c},$$
   where the last equality is due to the homogeneity of the policy, $\pi_{c\theta}(s) = \arg\max_{a \in \mathcal{A}} e_s^T c\theta_a = \arg\max_{a \in \mathcal{A}} e_s^T \theta_a$. By taking the limit, we have
$$\lim_{c \to \infty} \frac{f(c\theta)}{c} = (\gamma DP\Pi_{\pi_\theta} - D)\theta + \lim_{c \to \infty} \frac{\gamma DP(\Pi_{\pi_\theta} - \Pi_{\pi_{Q^*}})Q^*}{c}$$
$$= (\gamma DP\Pi_{\pi_\theta} - D)\theta = f_\infty(\theta).$$
   Moreover, $f$ is globally Lipschitz continuous according to Proposition 1. Therefore, the proof is completed.

2. The second statement of Assumption 1 follows from Lemma 5..

3. The third statement of Assumption 1 follows from Theorem 1.

4. Next, we prove the remaining parts. Recall that the Q-learning update is expressed as
$$Q_{k+1} = Q_k + \alpha_k(f(Q_k) + \varepsilon_{k+1})$$
   with the stochastic error
$$\varepsilon_{k+1} = (e_a \otimes e_s)(e_a \otimes e_s)^T R + \gamma(e_a \otimes e_s)(e_{s'})^T \Pi_{\pi_{Q_k}} Q_k$$
$$- (e_a \otimes e_s)(e_a \otimes e_s)^T Q_k - (DR + \gamma DP\Pi_{\pi_{Q_k}} Q_k - DQ_k)$$
   and
$$f(Q) = (\gamma DP\Pi_{\pi_Q} - D)Q + \gamma DP(\Pi_{\pi_Q} - \Pi_{\pi_{Q^*}})Q^*.$$
   Define the history $\mathcal{G}_k := (\varepsilon_k, \varepsilon_{k-1}, \ldots, \varepsilon_1, Q_k, Q_{k-1}, \ldots, Q_0)$, and the process $(M_k)_{k=0}^\infty$ with $M_k := \sum_{i=1}^k \varepsilon_i$. Then, we can prove that $(M_k)_{k=0}^\infty$ is Martingale. To do so, we first

prove $\mathbb{E}[\varepsilon_{k+1}|\mathcal{G}_k] = 0$ by

$$
\begin{aligned}
\mathbb{E}[\varepsilon_{k+1}|\mathcal{G}_k] =& \mathbb{E}[(e_a \otimes e_s)(e_a \otimes e_s)^T R|\mathcal{G}_k] + \mathbb{E}[\gamma(e_a \otimes e_s)(e_{s'})^T \Pi_{\pi_{Q_k}} Q_k|\mathcal{G}_k] \\
& - \mathbb{E}[(e_a \otimes e_s)(e_a \otimes e_s)^T Q_k|\mathcal{G}_k] - \mathbb{E}[DR + \gamma DP\Pi_{\pi_{Q_k}} Q_k - DQ_k|\mathcal{G}_k] \\
=& \mathbb{E}[DR + \gamma DP\Pi_{\pi_{Q_k}} Q_k - DQ_k|\mathcal{G}_k] - \mathbb{E}[DR + \gamma DP\Pi_{\pi_{Q_k}} Q_k - DQ_k|\mathcal{G}_k] \\
=& 0,
\end{aligned}
$$

where the second equality is due to the i.i.d. assumption of samples. Using this identity, we have

$$
\begin{aligned}
\mathbb{E}[M_{k+1}|\mathcal{G}_k] =& \mathbb{E}\left[\sum_{i=1}^{k+1} \varepsilon_i \,\middle|\, \mathcal{G}_k\right] = \mathbb{E}[\varepsilon_{k+1}|\mathcal{G}_k] + \mathbb{E}\left[\sum_{i=1}^{k} \varepsilon_i \,\middle|\, \mathcal{G}_k\right] \\
=& \mathbb{E}\left[\sum_{i=1}^{k} \varepsilon_i \,\middle|\, \mathcal{G}_k\right] = \sum_{i=1}^{k} \varepsilon_i = M_k.
\end{aligned}
$$

Therefore, $(M_k)_{k=0}^{\infty}$ is a Martingale sequence, and $\varepsilon_{k+1} = M_{k+1} - M_k$ is a Martingale difference. Moreover, it can be easily proved that the fourth condition of Assumption 1 is satisfied by algebraic calculations. Therefore, the fourth condition is met.

$\square$

## F  Proof of Theorem 3

*Proof.* Using $\gamma DP(\Pi_{\pi_{Q_t^B}} - \Pi_{\pi_{Q^*}})Q^* \leq 0$, we obtain

$$
\begin{aligned}
\begin{bmatrix} -D & \gamma DP\Pi_{\pi_{Q_t^B}} \\ \delta I & -\delta I \end{bmatrix} \begin{bmatrix} Q_t^A - Q^* \\ Q_t^B - Q^* \end{bmatrix} &+ \begin{bmatrix} \gamma DP(\Pi_{\pi_{Q_t^B}} - \Pi_{\pi_{Q^*}})Q^* \\ 0 \end{bmatrix} \leq \begin{bmatrix} -D & \gamma DP\Pi_{\pi_{Q_t^B}} \\ \delta I & -\delta I \end{bmatrix} \begin{bmatrix} Q_t^A - Q^* \\ Q_t^B - Q^* \end{bmatrix} \\
&\leq \begin{bmatrix} -D & \gamma DP\Pi_{\pi_{Q_t^B - Q^*}} \\ \delta I & -\delta I \end{bmatrix} \begin{bmatrix} Q_t^A - Q^* \\ Q_t^B - Q^* \end{bmatrix}.
\end{aligned}
$$

Consider the upper comparison system

$$
\frac{d}{dt}\begin{bmatrix} Q_t^{A,u} - Q^* \\ Q_t^{B,u} - Q^* \end{bmatrix} = \begin{bmatrix} -D & \gamma DP\Pi_{\pi_{Q_t^{B,u} - Q^*}} \\ \delta I & -\delta I \end{bmatrix} \begin{bmatrix} Q_t^{A,u} - Q^* \\ Q_t^{B,u} - Q^* \end{bmatrix}, \quad \begin{bmatrix} Q_0^{A,u} - Q^* \\ Q_0^{B,u} - Q^* \end{bmatrix} > \begin{bmatrix} Q_0^A - Q^* \\ Q_0^B - Q^* \end{bmatrix} \in \mathbb{R}^{2|\mathcal{S}||\mathcal{A}|},
$$

and define the vector functions

$$
\overline{f}(y_1, y_2) := \begin{bmatrix} \overline{f}_1(y_1, y_2) \\ \overline{f}_2(y_1, y_2) \end{bmatrix} := \begin{bmatrix} -D & \gamma DP\Pi_{\pi_{y_2}} \\ \delta I & -\delta I \end{bmatrix} \begin{bmatrix} y_1 \\ y_2 \end{bmatrix}
$$

$$
\underline{f}(y_1, y_2) := \begin{bmatrix} \underline{f}_1(z_1, z_2) \\ \underline{f}_2(z_1, z_2) \end{bmatrix} := \begin{bmatrix} -D & \gamma DP\Pi_{\pi_{z_2 + Q^*}} \\ \delta I & -\delta I \end{bmatrix} \begin{bmatrix} z_1 \\ z_2 \end{bmatrix} + \begin{bmatrix} \gamma DP(\Pi_{\pi_{z_2 + Q^*}} - \Pi_{\pi_{Q^*}})Q^* \\ 0 \end{bmatrix},
$$

and consider the systems

$$
\frac{d}{dt}\begin{bmatrix} y_{t,1} \\ y_{t,2} \end{bmatrix} = \begin{bmatrix} \overline{f}_1(y_{t,1}, y_{t,2}) \\ \overline{f}_2(y_{t,1}, y_{t,2}) \end{bmatrix}, \quad y_0 > \begin{bmatrix} Q_0^A - Q^* \\ Q_0^B - Q^* \end{bmatrix},
$$

$$
\frac{d}{dt}\begin{bmatrix} z_{t,1} \\ z_{t,2} \end{bmatrix} = \begin{bmatrix} \underline{f}_1(z_{t,1}, z_{t,2}) \\ \underline{f}_2(z_{t,1}, z_{t,2}) \end{bmatrix}, \quad z_0 = \begin{bmatrix} Q_0^A - Q^* \\ Q_0^B - Q^* \end{bmatrix},
$$

for all $t \geq 0$. We first prove that $\overline{f}$ is quasi-monotone increasing. We will check the condition of the quasi-monotone increasing function for $\overline{f}_1$ and $\overline{f}_2$, separately. Assume that $p_1 \in \mathbb{R}^{|\mathcal{S}||\mathcal{A}|}$ and $p_2 \in \mathbb{R}^{|\mathcal{S}||\mathcal{A}|}$ are nonnegative vectors, and an $i$the element of $p_1$ is zero. For $\overline{f}_1$, we have

$$
\begin{aligned}
e_i^T \overline{f}_1(y_1 + p_1, y_2 + p_2) =& - e_i^T D(y_1 + p_1) + \gamma e_i^T DP\Pi_{\pi_{(y_2 + p_2)}}(y_2 + p_2) \\
=& - e_i^T Dy_1 + \gamma e_i^T DP\Pi_{\pi_{(y_2 + p_2)}}(y_2 + p_2) \\
\geq& - e_i^T Dy_1 + \gamma e_i^T DP\Pi_{\pi_{y_2}} y_2 \\
=& e_i^T \overline{f}_1(y_1, y_2),
\end{aligned}
$$

where the second line is due to $-e_i^T D p_1 = 0$. Similarly, assuming that $p_1 \in \mathbb{R}^{|\mathcal{S}||\mathcal{A}|}$ and $p_2 \in \mathbb{R}^{|\mathcal{S}||\mathcal{A}|}$ are nonnegative vectors, and an $i$the element of $p_2$ is zero, we get

$$
\begin{aligned}
e_i^T \overline{f}_2(y_1 + p_1, y_2 + p_2) &= \delta e_i^T(y_1 + p_1) - \gamma \delta e_i^T(y_2 + p_2) \\
&= \delta e_i^T(y_1 + p_1) - \gamma \delta e_i^T y_2 \\
&\geq \delta e_i^T y_1 - \gamma \delta e_i^T y_2 \\
&= e_i^T \overline{f}_2(y_1, y_2),
\end{aligned}
$$

where the second line is due to $e_i^T p_2 = 0$. Therefore, $\overline{f}$ is quasi-monotone increasing. The Lipschitz continuity of $\overline{f}$ and $\underline{f}$ can be easily proved. Therefore, by Lemma 2, $\begin{bmatrix} Q_t^A - Q^* \\ Q_t^B - Q^* \end{bmatrix} \leq \begin{bmatrix} Q_t^{A,u} - Q^* \\ Q_t^{B,u} - Q^* \end{bmatrix}$

holds for all $t \geq 0$, where $\begin{bmatrix} Q_t^{A,u} - Q^* \\ Q_t^{B,u} - Q^* \end{bmatrix}$ is the solution of the upper comparison system.

Moreover, using the inequality $\gamma D P \Pi_{\pi_{Q_t^B}} Q_t^B \geq \gamma D P \Pi_{\pi_{Q^*}} Q_t^B$, we obtain

$$
\frac{d}{dt} \begin{bmatrix} Q_t^A \\ Q_t^B \end{bmatrix} = \begin{bmatrix} -D & \gamma D P \Pi_{\pi_{Q_t^B}} \\ \delta I & -\delta I \end{bmatrix} \begin{bmatrix} Q_t^A \\ Q_t^B \end{bmatrix} + \begin{bmatrix} DR \\ 0 \end{bmatrix} \geq \begin{bmatrix} -D & \gamma D P \Pi_{\pi_{Q^*}} \\ \delta I & -\delta I \end{bmatrix} \begin{bmatrix} Q_t^A \\ Q_t^B \end{bmatrix} + \begin{bmatrix} DR \\ 0 \end{bmatrix}.
$$

Using this relation, consider the lower comparison system

$$
\frac{d}{dt} \begin{bmatrix} Q_t^{A,l} - Q^* \\ Q_t^{B,l} - Q^* \end{bmatrix} = \begin{bmatrix} -D & \gamma D P \Pi_{\pi_{Q^*}} \\ \delta I & -\delta I \end{bmatrix} \begin{bmatrix} Q_t^{A,l} - Q^* \\ Q_t^{B,l} - Q^* \end{bmatrix}, \quad \begin{bmatrix} Q_0^{A,l} - Q^* \\ Q_0^{B,l} - Q^* \end{bmatrix} < \begin{bmatrix} Q_0^A - Q^* \\ Q_0^B - Q^* \end{bmatrix} \in \mathbb{R}^{2|\mathcal{S}||\mathcal{A}|},
$$

or equivalently,

$$
\frac{d}{dt} \begin{bmatrix} Q_t^{A,l} \\ Q_t^{B,l} \end{bmatrix} = \begin{bmatrix} -D & \gamma D P \Pi_{\pi_{Q^*}} \\ \delta I & -\delta I \end{bmatrix} \begin{bmatrix} Q_t^{A,l} \\ Q_t^{B,l} \end{bmatrix} + \begin{bmatrix} DR \\ 0 \end{bmatrix}.
$$

To proceed, define the vector functions

$$
\overline{f}(y_1, y_2) := \begin{bmatrix} \overline{f}_1(y_1, y_2) \\ \overline{f}_2(y_1, y_2) \end{bmatrix} = \begin{bmatrix} \gamma D P \Pi_{\pi_{y_2}} & -D \\ \delta I & -\delta I \end{bmatrix} \begin{bmatrix} y_1 \\ y_2 \end{bmatrix} + \begin{bmatrix} DR \\ 0 \end{bmatrix}
$$

$$
\underline{f}(z_1, z_2) := \begin{bmatrix} \underline{f}_1(z_1, z_2) \\ \underline{f}_2(z_1, z_2) \end{bmatrix} = \begin{bmatrix} -D & \gamma D P \Pi_{\pi_{Q^*}} \\ \delta I & -\delta I \end{bmatrix} \begin{bmatrix} z_1 \\ z_2 \end{bmatrix} + \begin{bmatrix} DR \\ 0 \end{bmatrix},
$$

and consider the systems

$$
\frac{d}{dt} \begin{bmatrix} y_{t,1} \\ y_{t,2} \end{bmatrix} = \begin{bmatrix} \overline{f}_1(y_{t,1}, y_{t,2}) \\ \overline{f}_2(y_{t,1}, y_{t,2}) \end{bmatrix}, \quad y_0 = \begin{bmatrix} Q_0^A - Q^* \\ Q_0^B - Q^* \end{bmatrix},
$$

$$
\frac{d}{dt} \begin{bmatrix} z_{t,1} \\ z_{t,2} \end{bmatrix} = \begin{bmatrix} \underline{f}_1(z_{t,1}, z_{t,2}) \\ \underline{f}_2(z_{t,1}, z_{t,2}) \end{bmatrix}, \quad z_0 < \begin{bmatrix} Q_0^A - Q^* \\ Q_0^B - Q^* \end{bmatrix},
$$

for all $t \geq 0$. Similar to the upper comparison systems, we can easily prove that $\overline{f}$ is quasi-monotone increasing, $\overline{f}$ and $\underline{f}$ are Lipschitz continuous. Therefore, applying similar steps as before and using Lemma 2, we have that $\begin{bmatrix} Q_t^A - Q^* \\ Q_t^B - Q^* \end{bmatrix} \geq \begin{bmatrix} Q_t^{A,l} - Q^* \\ Q_t^{B,l} - Q^* \end{bmatrix}$ holds for all $t \geq 0$, where $\begin{bmatrix} Q_t^{A,l} - Q^* \\ Q_t^{B,l} - Q^* \end{bmatrix}$ is the solution of the linear system

Now, it remains to prove the asymptotic convergence of the comparison systems. For notational convenience, we define $\Pi_\sigma, \sigma \in \mathcal{M}$ as $\Pi_{\pi_{Q_t^B}}$ such that $\sigma = \psi(\pi_{Q_t^B})$. Then, for the upper comparison switching system, we apply Lemma 3 with $A_\sigma = \begin{bmatrix} -D & \gamma D P \Pi_\sigma \\ \delta I & -\delta I \end{bmatrix}$ and $L = \begin{bmatrix} I & 0 \\ 0 & \gamma^{1/2} I \end{bmatrix}$, which

satisfies $L A_\sigma = \bar{A}_\sigma L$ with $\bar{A}_\sigma = \begin{bmatrix} -D & \gamma^{1/2} D P \Pi_\sigma \\ \gamma^{1/2} \delta I & -\delta I \end{bmatrix}$. To check the strictly negative row

dominating diagonal condition, for $i \in \{1, 2, \ldots, |\mathcal{S}||\mathcal{A}|\}$, we have

$$[\bar{A}_\sigma]_{ii} + \sum_{j \in \{1,2,\ldots,n\}\setminus\{i\}} |[\bar{A}_\sigma]_{ij}| = [-D]_{ii} + \gamma^{1/2}[-D]_{ii} \sum_{j \in \{1,2,\ldots,n\}\setminus\{i\}} |[P\Pi_\sigma]_{ij}|$$

$$\leq [-D]_{ii} + \gamma^{1/2}[-D]_{ii}$$

$$\leq -1 + \gamma^{1/2} < 0.$$

For $i \in \{|\mathcal{S}||\mathcal{A}| + 1, |\mathcal{S}||\mathcal{A}| + 2, \ldots, 2|\mathcal{S}||\mathcal{A}|\}$, we also have

$$[\bar{A}_\sigma]_{ii} + \sum_{j \in \{1,2,\ldots,n\}\setminus\{i\}} |[\bar{A}_\sigma]_{ij}| = -\delta + \delta\gamma^{1/2} = \delta(-1 + \gamma^{1/2}) < 0$$

for any $\delta > 0$. Therefore, the strictly negative row dominating diagonal condition is satisfied. By Lemma 3, the origin of the switching system (15) is globally asymptotically stable. The lower comparison system's stability can be proved in an equivalent way. Since the switching system's solution is upper and lower bounded by the corresponding comparison systems, it asymptotically converges to the origin. This completes the proof. $\qquad\square$

## G  Proof of Theorem 5

*Proof.* By Assumption 2, it holds that $\gamma\Phi^T DP(\Pi_{\pi_{\Phi\theta_t^l}} - \Pi_{\pi_{\Phi\theta^*}})\Phi\theta^* \leq 0$, $\Phi^T(\gamma DP\Pi_{\pi_{\Phi\theta_t^u}} - D)\Phi(\theta_t^u - \theta^*) \leq \Phi^T(\gamma DP\Pi_{\pi_{\Phi(\theta_t^u - \theta^*)}} - D)\Phi(\theta_t^u - \theta^*)$, and we obtain the upper comparison system

$$\frac{d}{dt}(\theta_t^u - \theta^*) = \Phi^T(\gamma DP\Pi_{\pi_{\Phi(\theta_t^u - \theta^*)}} - D)\Phi(\theta_t^u - \theta^*),$$
$$\theta_0^u - \theta^* > \theta_0 - \theta^* \in \mathbb{R}^n. \tag{22}$$

To proceed, define the vector functions

$$\overline{f}(y) = \Phi^T(\gamma DP\Pi_{\pi_{\Phi y}} - D)\Phi y$$
$$\underline{f}(z) = (\gamma\Phi^T DP\Pi_{\pi_{\Phi(z+\theta^*)}}\Phi - \Phi^T D\Phi)z + \gamma\Phi^T DP(\Pi_{\pi_{\Phi(z+\theta^*)}} - \Pi_{\pi_{\Phi\theta^*}})\Phi\theta^*, ,$$

and consider the systems

$$\frac{d}{dt}y_t = \overline{f}(y_t), \quad y_0 > \theta_0 - \theta^*,$$
$$\frac{d}{dt}z_t = \underline{f}(z_t), \quad z_0 = \theta_0 - \theta^*,$$

for all $t \geq 0$. To apply Lemma 3, we first check the quasi-monotonicity of $\overline{f}$. For any nonnegative vector $p$ such that its $i$th element is zero, we have

$$e_i^T \overline{f}(y + p) = e_i^T(\gamma\Phi^T DP\Pi_{\Phi(y+p)}\Phi - \Phi^T D\Phi)(y + p)$$
$$= \gamma e_i^T \Phi^T DP\Pi_{\Phi(y+p)}\Phi(y + p) - e_i^T \Phi^T D\Phi p - e_i^T \Phi^T D\Phi y$$
$$= \gamma e_i^T \Phi^T DP\Pi_{\Phi(y+p)}\Phi(y + p) - e_i^T \Phi^T D\Phi y$$
$$= \gamma e_i^T \Phi^T DP \begin{bmatrix} \max_a(\Phi(y + p))_a(1) \\ \max_a(\Phi(y + p))_a(2) \\ \vdots \\ \max_a(\Phi(y + p))_a(|\mathcal{S}|) \end{bmatrix} - e_i^T \Phi^T D\Phi y$$
$$\geq \gamma e_i^T \Phi^T DP \begin{bmatrix} \max_a(\Phi(y))_a(1) \\ \max_a(\Phi(y))_a(2) \\ \vdots \\ \max_a(\Phi(y))_a(|\mathcal{S}|) \end{bmatrix} - e_i^T \Phi^T D\Phi y$$
$$= \gamma e_i^T \Phi^T DP\Pi_{\Phi(y)}\Phi(y) - e_i^T \Phi^T D\Phi y$$
$$= e_i^T \overline{f}(y),$$

where the third line is due to Assumption 3 and the fact that $\Phi^T D\Phi$ is a diagonal matrix. Therefore, $\overline{f}$ is quasi-monotone increasing. The Lipschitz continuity of $\overline{f}$ and $\underline{f}$ can be provided following similar

lines of the proof of Proposition 1, where we can use the fact that $f(z) = (\gamma DP\Pi_{\pi_{(z+Q^*)}} - D)(z + Q^*) + DR$. Therefore, the vector comparison principle, Lemma 2, leads to $\theta_t \leq \theta_t^u$ as $t \to \infty$.

On the other hand, by Assumption 2, it holds that $\gamma\Phi^T DP\Pi_{\pi_{\Phi\theta_t}}\Phi\theta_t \geq \gamma\Phi^T DP\Pi_{\pi_{\Phi\theta^*}}\Phi\theta_t$, and we obtain the lower comparison system

$$\frac{d}{dt}(\theta_t^l - \theta^*) = \Phi^T(\gamma DP\Pi_{\pi_{\Phi\theta^*}} - D)\Phi(\theta_t^l - \theta^*),$$
$$\theta_0^l - \theta^* < \theta_0 - \theta^* \in \mathbb{R}^n,$$

or equivalently,

$$\frac{d}{dt}\theta_t^l = \Phi^T(\gamma DP\Pi_{\pi_{\Phi\theta^*}} - D)\Phi\theta_t^l - \Phi^T(\gamma DP\Pi_{\pi_{\Phi\theta^*}} - D)\Phi\theta^*,$$
$$\theta_0^l < \theta_0 \in \mathbb{R}^n.$$

To proceed, define the vector functions

$$\overline{f}(y) = \Phi^T(\gamma DP\Pi_{\pi_{\Phi y}} - D)\Phi y + \Phi^T DR$$
$$\underline{f}(z) = \Phi^T(\gamma DP\Pi_{\pi_{\Phi\theta^*}} - D)\Phi z - \Phi^T(\gamma DP\Pi_{\pi_{\Phi\theta^*}} - D)\Phi\theta^*,$$

and consider the systems

$$\frac{d}{dt}y_t = \overline{f}(y_t), \quad y_0 = \theta_0,$$
$$\frac{d}{dt}z_t = \underline{f}(z_t), \quad z_0 < \theta_0,$$

for all $t \geq 0$. To apply Lemma 3, we check the quasi-monotonicity of $\overline{f}$, which can be easily proved following the steps for the upper comparison system. The Lipschitz continuity of $\overline{f}$ and $\underline{f}$ can be also proved following similar lines of the proof of Proposition 1. Therefore, Lemma 2 leads to $\theta_t \geq \theta_t^l$ as $t \to \infty$.

To prove the asymptotic stability of the original system (18), it is sufficient to prove that the upper and lower comparison systems are globally asymptotically stable. In this respect, we can apply Lemma 3 to obtain a sufficient condition for the stability. In particular, both the upper and lower comparison systems are globally asymptotically stable if the switching system is globally asymptotically stable

$$\frac{d}{dt}\theta_t = A_{\sigma_t}\theta_t,$$

under arbitrary switchings, $\sigma_t$, where $A_{\psi(\pi)} = \Phi^T(\gamma DP\Pi_{\psi(\pi)} - D)\Phi$ for all $\pi \in \Theta_\Phi$. By Lemma 3, it is true if and only if

$$[A_{\psi(\pi)}]_{ii} + \sum_{j \in \{1,2,\ldots,n\}\setminus\{i\}} |[A_{\psi(\pi)}]_{ij}|$$

$$= [\Phi^T(\gamma DP\Pi_{\psi(\pi)} - D)\Phi]_{ii} + \sum_{j \in \{1,2,\ldots,n\}\setminus\{i\}} |[\Phi^T(\gamma DP\Pi_{\psi(\pi)} - D)\Phi]_{ij}|$$

$$= \phi_i^T(\gamma DP\Pi_{\psi(\pi)})\phi_i - \phi_i^T D\phi_i + \sum_{j \in \{1,2,\ldots,n\}\setminus\{i\}} \phi_i^T(\gamma DP\Pi_{\psi(\pi)} - D)\phi_j$$

$$= -\phi_i^T D\phi_i + \sum_{j \in \{1,2,\ldots,n\}} \phi_i^T \gamma DP\Pi_{\psi(\pi)}\phi_j$$

$$= -\phi_i^T D\phi_i + \phi_i^T \gamma DP\Pi_{\psi(\pi)} \sum_{j \in \{1,2,\ldots,n\}} \phi_j$$

$$< 0$$

for all $i \in \{1, 2, \ldots, n\}, \pi \in \Theta_\Phi$, where the second line is due to Assumption 2, and the fourth line is due to Assumption 3 and the fact that $\phi_i^T D\phi_j = 0$ for $j \neq i$. This completes the proof. $\qquad\square$

# H    Proof of Proposition 3

*Proof.* If the elements of the feature matrix $\Phi$ are binary numbers, then since the columns of $\Phi$ consist of sums of distinct basis vectors, $e_i \in \mathbb{R}^{|\mathcal{S}||\mathcal{A}|}$, and it follows that

$$\sum_{j \in \{1,2,\dots,n\}} \phi_j \leq \mathbf{1}_{|\mathcal{S}||\mathcal{A}|}, \tag{23}$$

where $\mathbf{1}_{|\mathcal{S}||\mathcal{A}|}$ is the vector with all elements being ones. The right-hand side of the condition in Theorem 6 is bounded as

$$\begin{aligned}
[A_{\psi(\pi)}]_{ii} + \sum_{j \in \{1,2,\dots,n\}\setminus\{i\}} |[A_{\psi(\pi)}]_{ij}| &= -\phi_i^T D\phi_i + \phi_i^T \gamma DP\Pi_{\psi(\pi)} \sum_{j \in \{1,2,\dots,n\}} \phi_j \\
&\leq -\phi_i^T D\phi_i + \phi_i^T \gamma DP\Pi_{\psi(\pi)} \mathbf{1}_{|\mathcal{S}||\mathcal{A}|} \\
&= -\phi_i^T D\phi_i + \gamma \phi_i^T D\mathbf{1}_{|\mathcal{S}||\mathcal{A}|} \\
&= -\phi_i^T D\phi_i + \gamma \phi_i^T D\phi_i \\
&= (\gamma - 1)\phi_i^T D\phi_i \\
&< 0,
\end{aligned}$$

where the first line comes from Theorem 6, the second line is due to (23), and the third line is due to the fact that $P\Pi_{\psi(\pi)}$ is a stochastic matrix, i.e., its low sums are one. This completes the proof. $\square$

# I    Proof of Proposition 2

*Proof.* The basic idea of the proof relies on the fact that Melo's sufficient condition ensures the existence of a quadratic Lyapunov function for the upper comparison system (22) following the results in [24]. Since the new sufficient condition, Proposition 2, is a necessary and sufficient condition for the global asymptotic stability of the upper comparison system, the Melo's condition implies the proposed new condition. Suppose that Melo's sufficient condition holds, and consider the quadratic Lyapunov function candidate: $V(\theta_t - \theta^*) := \frac{1}{2}(\theta_t - \theta^*)^T(\theta_t - \theta^*)$.

Its time derivative along the state trajectories of the upper comparison system (22) is given by

$$\begin{aligned}
\frac{d}{dt}V(\theta_t - \theta^*) &= (\theta_t - \theta^*)^T \Phi^T(\gamma DP\Pi_{\pi_{\Phi\theta_t}} - D)\Phi(\theta_t - \theta^*) \\
&= \gamma(\theta_t - \theta^*)^T \Phi^T DP\Pi_{\pi_{\Phi\theta_t}} \Phi(\theta_t - \theta^*) - (\theta_t - \theta^*)^T \Phi^T D\Phi(\theta_t - \theta^*) \\
&= -(\theta_t - \theta^*)^T \Phi^T D\Phi(\theta_t - \theta^*) + \gamma\mathbb{E}[(\theta_t - \theta^*)^T \Phi^T(e_a \otimes e_s)(e_{s'})^T \Pi_{\pi_{\Phi\theta_t}} \Phi(\theta_t - \theta^*)\},
\end{aligned}$$

where $(s, a)$ is sampled from the stationary state-action distribution and $s' \sim P_a(s, \cdot)$. Similar to the ideas in [24], using Holder's inequality leads to

$$\begin{aligned}
&\frac{d}{dt}V(\theta_t - \theta^*) \\
&= (\theta_t - \theta^*)^T \Phi^T(\gamma DP\Pi_{\pi_{\Phi\theta_t}} - D)\Phi(\theta_t - \theta^*) \\
&\leq -(\theta_t - \theta^*)^T \Phi^T D\Phi(\theta_t - \theta^*) + \gamma\sqrt{\mathbb{E}[(\theta_t - \theta^*)^T \Phi^T(e_a \otimes e_s)(e_a \otimes e_s)^T \Phi(\theta_t - \theta^*)]} \\
&\quad \times \sqrt{\mathbb{E}[(\theta_t - \theta^*)^T \Phi^T \Pi_{\pi_{\Phi\theta_t}}^T(e_{s'})(e_{s'})^T \Pi_{\pi_{\Phi\theta_t}} \Phi(\theta_t - \theta^*)]} \\
&= -(\theta_t - \theta^*)^T \Phi^T D\Phi(\theta_t - \theta^*) + \gamma\sqrt{(\theta_t - \theta^*)^T \Phi^T D\Phi(\theta_t - \theta^*)} \times \sqrt{(\theta_t - \theta^*)^T \Phi^T \Pi_{\pi_{\Phi\theta_t}}^T D^\beta \Pi_{\pi_{\Phi\theta_t}} \Phi(\theta_t - \theta^*)},
\end{aligned}$$

where the last equality uses the fact that the distribution of $s'$ is identical to the distribution of $s$. Now, we apply the Melo's condition to have

$$\begin{aligned}
&\frac{d}{dt}V(\theta_t - \theta^*) \\
&< -(\theta_t - \theta^*)^T \Phi^T D\Phi(\theta_t - \theta^*) + \gamma\sqrt{(\theta_t - \theta^*)^T \Phi^T D\Phi(\theta_t - \theta^*)}\sqrt{\frac{1}{\gamma^2}(\theta_t - \theta^*)^T \Phi^T D\Phi(\theta_t - \theta^*)} \\
&= -(\theta_t - \theta^*)^T \Phi^T D\Phi(\theta_t - \theta^*) + (\theta_t - \theta^*)^T \Phi^T D\Phi(\theta_t - \theta^*) \\
&= 0, \quad \forall \theta_t - \theta^* \neq 0.
\end{aligned}$$

This implies that $V$ is a Lyapunov function. By the standard Lyapunov theorem, the origin of the upper comparison system (22) is globally asymptotically stable. The proof holds even if the upper comparison system is arbitrarily switching. Since the new sufficient condition in Proposition 2 is a necessary and sufficient condition for the global asymptotic stability of the upper comparison system (22) under arbitrary switching, this implies that the proposed new condition holds. □