[Reviews · NeurIPS 2020]

Review 1

Summary and Contributions: The paper studies the convergence of various Q-learning methods in finite MDPs. Its main idea is to show that Q-learning can be reformulated as an affine switching system which itself can be analyzed based on two linear switching systems (an upper and a lower comparison system). This reformulation makes the rich theory of linear switching systems (developed in control theory) applicable to variants of Q-learning. The authors use this methodology to provide new convergence theorems not only for the standard (asynchronous) Q-learning, but also for a recently proposed averaging variant (for which no convergence results existed previously). New sufficient conditions are given for the convergence of Q-learning with linear function approximation (for finite MDPs), as well.

Strengths: - The paper is well-written and relatively easy to follow. The ideas are well explained. - The results are significant: the connection between the theories of Q-learning and switching linear systems could lead to new developments in RL. - New convergence theorems are given for Q-learning, averaging Q-learning, and Q-learning with linear function approximation. - The paper also has an educational value, as it nicely overviews the related techniques.

Weaknesses: - Only finite MDPs are studies and these ideas cannot be easily transferred to MDPs with more general state and action spaces, such as Borel MDPs. Studying Q-learning with linear function approximation is more important for continuous state spaces. - In Section 3 the observed state-action trajectory is assumed to be i.i.d. which is unrealistic and does not hold in most RL applications. - If I understand correctly, the step-sizes are assumed to be independent of the visited state-action pairs, which is also not an innocent assumption as in practice the learning rates typically do depend on the observations and such a dependence is very important. - Assumption 3 (page 7) is restrictive, as it does not hold for the typical choices of feature functions. - The paper does not contain any experimental evaluations. - The provided theorems are all asymptotic. No finite sample guarantees are given.

Correctness: The theoretical claims are correct. The paper does not contain any empirical evaluations.

Clarity: The paper is well-written.

Relation to Prior Work: The previous works are adequately discussed.

Reproducibility: Yes

Additional Feedback: Minor comments: - Page1, line 12: "Watkin" should be "Watkins". - Page1, line 21: Q-learning is also a variant of TD-learning. - Page 7, line 248: the reference is missing.


Review 2

Summary and Contributions: The paper proposes a unified method for the analysis of stability and convergence of several Q-learning variants using principles from switching and stochastic systems. The paper is mainly mathematical and there are no examples to illustrate the theorems.

Strengths: The paper seems rigorous and the results seem strong. "UPDATE after Rebuttal" I am satisfied with the author response and the small example. It would also be interesting to see a larger example and to understand what happens with respect to the assumptions. I encourage this to be done in the final version. I maintain my positive score.

Weaknesses: The paper consists of only maths. It would be beneficial to illustrate the results, for instance by simulating the upper and lower bounding systems used in the proof. Also, it is not clear what the implications of the results are as well as their assumptions. A discussion section should be added where the maths is "demystified" and the assumptions, scope and limitations are made clear.

Correctness: The claims appear correct as the proofs are mathematically sound.

Clarity: The paper is not easyy to read due to the heavy notation and the amount of maths. It would be beneficial to work more on presentation and perhaps leave some of the notation overload to the appendix. In the main paper, the overall message should prevail over the line by line derivation of the results.

Relation to Prior Work: I think in the introduction it is made clear where this paper stands with respect to the cited literature.

Reproducibility: Yes

Additional Feedback:


Review 3

Summary and Contributions: This paper attempts to provide convergence results for variants of Q-learning, which are key algorithms for classical reinforcement learning, using stability analysis from nonlinear ordinary differential equations (ODE) theory and arguments from switching systems theory. While the convergence of Q-learning is well-known, the key contribution of this paper, in the view of the authors, is to extend convergence results to variants of Q-learning, such as average Q-learning, asynchronous Q-learning, and Q-learning with linear function approximation, under the unifying lens of switching systems. %%%%%%%%%%%%%%%%%%%% Post rebuttal: I would like to thank the authors for the clarifications they provided in their response. Unfortunately, my concern about the experimental evaluation still remains, and thus I will keep my score as is.

Strengths: The main technical argument of this paper is to view Q-learning updates as stochastic recursions in an ODE. Further, the authors provide a novel, sound contribution illustrating that the error between a Q-value approximation and the optimal Q-value converges to zero by showing convergence to a fixed point of an affine switching system that governs the error dynamics. The theme of this work is a clear fit for the NeurIPS community due to prevalent interest in guarantees on the convergence of RL algorithms.

Weaknesses: The main criticism I have with this paper is the lack of in-depth, illustrative numerical examples. The authors present an extremely simple MDP in Example 1 (Line 288) in order to illustrate that existing convergence bounds are too weak and fail to guarantee convergence for linear policy approximation, but the tighter bounds in this paper successfully do so. First, the writing from lines 291-296 is hard to understand and the bigger picture is missing. For example, how did the authors come up with the feature matrix values in line 291 and what do they represent in the context of this MDP? More broadly, Example 1 does not provide any sort of numerical evidence for the stability analyses claimed in this paper. Instead, the authors could actually show the numerical convergence of average Q-learning in a well-studied model dynamical system. Then, they can implement the solution to an ODE based on the switching systems central to this paper and show the error dynamics converge to zero (Equation 15). This would convincingly provide numerical evidence for the results from this paper.

Correctness: The theorems are correct and the appendix provides detailed, sound technical analysis.

Clarity: The paper is fairly well written, but is very dense to read since the reader is inundated with theorems without much motivation or introduction.

Relation to Prior Work: The authors provide an adequate survey of convergence results for Q-learning and do a good job of putting their contributions in perspective.

Reproducibility: Yes

Additional Feedback:


Review 4

Summary and Contributions: The goal of the paper is to prove (asymptotic) convergence of asynchronous Q-learning and other variants of Q-learning within a “simplified” common framework. This is done using the (commonly adopted) ODE method and the results in Borkar and Meyn, but crucially, modeling the ODEs as switched linear systems with state-dependent switching policies. This allows the authors to unify the treatment across different instances of Q-learning, and in particular establish convergence of some interesting variants of Q-learning (“averaging Q-learning” which essentially amounts to target tracking with a Polyak average; Q-learning with a linear state-space function approximator). The established theory of switching systems is a key tool in the paper, and its usage in the context of an analysis of RL algorithms may be of interest in its own right. The novelty of the paper stems principally from the use of switching systems as an analysis tool; the authors provide a condition for global asymptotic convergence of Q-learning with linear function approximators that appears to be weaker than in previously published work. The proof of convergence of averaging Q-learning is ‘novel’ insofar as the algorithm was only recently introduced. Nevertheless, that the framework readily applies to the analysis of this variant is noteworthy, as averaging Q-learning can be seen as representative of modifications of Q-learning a practitioner might be tempted to employ (e.g. averaging of iterates, target tracking).

Strengths: Borrowing techniques from switching systems theory appears to be novel, leads to simpler proofs of existing convergence results, and does appear to straightforwardly extend to variants of the vanilla Q-learning method while the more commonly known previous analyses may not be adapted quite as readily. A difficulty the authors encountered was that they needed to show global asymptotic stability of a switching affine system in order to apply existing results (Borkar and Meyn), and ultimately show convergence via the ODE method. This led to a creative comparison technique (the affine term is what poses the difficulty). Although one might criticize the paper for relying quite so heavily on pre-existing techniques (and closely following the ODE method as it’s essential tack), I see this as actually a strength; the ODE method is simple, compelling, and broadly applicable. In the current context, the use of the ODE approach leads to the introduction of an additional verification burden around the switching system, but it’s practically manageable as the example analyses demonstrate. Finally, the paper is relatively well organized, and the authors have struck a good balance in deciding what background or details to include and what to reference or relegate to the supplementary material.

Weaknesses: It should be emphasized that convergence of Q-learning, both asymptotic and non-asymptotic with rates, has been previously established; this paper’s main contribution is, in my mind, to simplify and unify. While the paper does a reasonably good job of achieving these goals, the work does feel slightly incremental. Perhaps a more minor point because it is purely conjecture: I’m not sure the synchronous/asynchronous Q-learning distinction (and novelty of the async. analysis) is as significant as argued -- I would expect that the method in [4,5] can probably be adjusted to accommodate asynchronous updates without tremendous effort. (Nevertheless, the approach here does simplify the analysis.) The analysis technique hinges on fairly strong global lipschitz continuity assumptions. If this could be relaxed, it may open the door to analysis of a wider class of algorithms. Finally, it would be helpful if the paper discussed limitations of the approach: not only where the method would obviously fail or become clearly non-applicable, but also (ideally common) instances where verifying the required conditions would become difficult, or intractable.

Correctness: Yes, the claims and general development appear to be sound. I did not, however, check the proofs in the supplementary material.

Clarity: Yes.

Relation to Prior Work: Yes

Reproducibility: Yes

Additional Feedback: Typos: Lemma 2, line 107, should read x_t instead of v_t. Eq (5), max_a Q_k(s_{k+1}, a) Question: Why is Lemma 3 needed vs. just requiring that the subsystem matrices have negative spectrum? Clearly Lemma 3 is weaker, but it comes at the expense of being a bit opaque. Update: I have read the rebuttal and thank the authors for their comments; I am satisfied with their responses to my questions.


Review 5

Summary and Contributions: The paper brings techniques from control theory to address stability and convergence of Q-learning with linear function approximation. The theoretical results offer an improvement on the sufficient condition of Melo et al.

Strengths: The set of tools for convergence analysis of RL is growing all the time, but remains limited. The paper brings a new approach, by applying theory of switched linear systems. This theory is all based on piecewise linear systems, so the authors required an extension to the piecewise affine functions found in Q-learning. This extension was made possible through the introduction of a bounding technique (the comparison theorem, Lemma 2) The application to Q-learning with function approximation is very nice, and great to finally see an improvement on the very crude ‘Melo’s condition’. The application to Watkins’ algorithm brings additional clarity to their approach.

Weaknesses: Lemma 3 on stability of switching systems gives sufficient conditions for stability in an adversarial setting—worst case over all switching signals. In the applications to RL of interest, switching is clearly state dependent. It is unfortunate that more structure cannot be exploited. I do understand that this is more a criticism of the state of the art of the general stability theory than a fault in their analysis. It is a pity that they could not find an algorithm for verification of (19), or some simplification. I did not find the discussion on averaging very satisfying. (13,14) is a in fact a crude variant of Ruppert-Polyak-Juditsky averaging. This should have been made clear, and it would have been much more exciting to see analysis of the real RPJ scheme. I don’t think it is entirely accurate to say that the Borkar-Meyn theorem only applies to the synchronous setting, even though this was the example considered. There is an SA interpretation of Watkins’ algorithm in the asynchronous setting in which the noise is martingale difference (note also that there are many follow up papers that extend the theory in various directions).

Correctness: I have read the paper carefully and did not find technical errors.

Clarity: It is very clearly written.

Relation to Prior Work: There is no serious relationship with either of the prior papers, "Asynchronous stochastic approximation and Q-learning" or "Error bounds for constant step-size Q-learning”. The clever monotonicity argument in this paper has roots in the 1997 paper of Szepesvari, "The asymptotic convergence-rate of Q-learning” (either this or a followup). This is an interesting historical note and not a criticism of the paper under review.

Reproducibility: Yes

Additional Feedback: I would be pleased to see the paper at NeurIPS since it brings new technical tools to RL, and clearly exposes a significant gap in the RL literature. It is definitely in the top 50% of papers that I have reviewed for this workshop.

[Author Response · NeurIPS 2020]

We thank all reviewers for their useful feedback and acknowledgement of our contribution. All comments will be
addressed in greater details in the revision. We first answer some common questions brought up by reviewers.

**Numerical illustrations**: Thanks for the suggestion! We agree with reviewers that it is useful to provide some numerical
evidence to illustrate the stability analysis. In Figures (a) and (b), we provide preliminary numerical illustrations of the
associated ODE models (original affine switching systems, upper and lower comparison systems) of the *asynchronous*
*Q-learning* and the *averaging Q-learning* on a toy MDP example with with $|S| = 2$ and $|A| = 2$. Our simulation
empirically verifies the theory claimed in the paper. Richer numerical evidence will be included in the revision.

(a) Stability of asynchronous Q-learning          (b) Stability of averaging Q-learning

**Assumptions**: Given that this is the first work that bridges switching system theory with RL algorithms, we intentionally
adopt simplified assumptions (such as i.i.d. assumption, orthogonal feature vectors) to avoid complications. Indeed,
these assumptions are quite common in many seminal work in RL theory and can be relaxed. We will dedicate a
discussion section on these assumptions and discuss potential relaxations or limitations.

Below we address the each reviewer's comments separately.

### Response to Reviewer 1

**Step-sizes**: Using learning rates dependent on state-action observations may be useful in practice for small tabular
MDPs; however, for modern developments in RL with function approximation, using learning rates independent of
state-action observations is dominant in the literature.

**Finite-sample guarantee**: Our current framework only provides asymptotic convergence similar as most work on ODE
analysis. Recent advances [Srikant & Ying, 2019; Hu & Syed, 2019; Chen et al., 2019; Wang & Giannakis, 2020;
Devraj & Meyn , 2020] show promise in the derivation of non-asymptotic convergence rates using more sophisticated
ODE analysis tools. We leave this extension for future investigation.

### Response to Reviewer 2
**Simulation/Assumptions**: See discussions above.
**Clarity of the paper**: We will improve the presentation of the paper and avoid heavy notations.

### Response to Reviewer 3
**Numerical evidence.** See discussions above.

**Example.** We used Example 1 as a *simple analytical illustration* of the tightness of these sufficient conditions. We
agree that it might be too simple to justify the claim. We will consider nontrivial examples with interpretable feature
matrices and verify these sufficient conditions numerically.

### Response to Reviewer 4

**Global Lipschitz continuity**: This is not necessarily an assumption. We show that all the ODE models associated with
Q-learning and its variants in this paper indeed satisfy the global Lipschitz continuity. Relaxing this condition into
weaker ones would be promising to accommodate a wider array of RL algorithms, which we will pursue in the future.

**Lemma 3**: The spectrum is not well defined for switching systems because the subsystem matrices change. It is known
that each subsystem matrix having negative spectrum does not guarantee the stability of the overall switching system.
Lemma 3 is a particular *necessary and sufficient condition* for the stability of the overall switching system.

[Meta-Review · NeurIPS 2020]

The paper provides a new tool (theory of switching systems) for the convergence analysis of RL algorithms that can be of interest to the wider RL theory community. Compared with existing results, several improvements are made. Authors should revise the paper to address reviewer comments. Prior works in this area need to be discussed more carefully, as pointed out by reviewers.